# Coordination of metal center biogenesis in human cytochrome c oxidase

Eva Nývltová[1], Jonathan V. Dietz[2], Javier Seravalli[3], Oleh Khalimonchuk [2,3] & Antoni Barrientos [1,4 ✉]

Mitochondrial cytochrome c oxidase (CcO) or respiratory chain complex IV is a heme $aa_3$-copper oxygen reductase containing metal centers essential for holo-complex biogenesis and enzymatic function that are assembled by subunit-specific metallochaperones. The enzyme has two copper sites located in the catalytic core subunits. The COX1 subunit harbors the $Cu_B$ site that tightly associates with heme $a_3$ while the COX2 subunit contains the binuclear $Cu_A$ site. Here, we report that in human cells the CcO copper chaperones form macromolecular assemblies and cooperate with several twin $CX_9C$ proteins to control heme $a$ biosynthesis and coordinate copper transfer sequentially to the $Cu_A$ and $Cu_B$ sites. These data on CcO illustrate a mechanism that regulates the biogenesis of macromolecular enzymatic assemblies with several catalytic metal redox centers and prevents the accumulation of cytotoxic reactive assembly intermediates.

[1] Department of Neurology, University of Miami Miller School of Medicine, 1420NW 9th Ave, Miami, FL 33136, USA. [2] Department of Biochemistry, University of Nebraska-Lincoln, 1901 Vine St. Beadle Center, Lincoln, NE 68588, USA. [3] Nebraska Redox Biology Center, University of Nebraska-Lincoln, 1901 Vine St. Beadle Center, Lincoln, NE 68588, USA. [4] Department of Biochemistry and Molecular Biology, University of Miami Miller School of Medicine, 1420NW 9th Ave, Miami, FL 33136, USA. ✉email: abarrientos@med.miami.edu

A erobic life relies on electron transport reactions that facilitate fundamental processes such as cellular respiration[1]. These biological electron transfer reactions are performed predominantly by proteins and rely on the presence of critical redox cofactors such as heme $a$ and copper (Cu), the prosthetic groups of cytochrome $c$ oxidase (CcO)[2]. However, for enzymatic complexes with multiple redox cofactors, uncoordinated or dysregulated biogenesis of the redox centers may result in the overproduction of reactive intermediates leading to progressive loss of cellular function and eventual death[3,4]. Therefore, cells must develop safety mechanisms to coordinate the biogenesis of multimeric enzymes that contain multiple redox centers. In most systems, such as mammalian CcO, these mechanisms remain incompletely characterized.

CcO is the terminal complex of the electron transport chain (ETC), required for aerobic adenosine 5'-triphosphate (ATP) production from aerobic bacteria to mammals[2]. Mammalian CcO or mitochondrial ETC complex IV (CIV) is formed by three conserved catalytic core transmembrane subunits (COX1, COX2, and COX3) encoded by the mitochondrial genome (mtDNA) and eleven nucleus-encoded accessory subunits[2]. CcO has two copper (Cu)-centers: Cu$_A$ in COX2 and Cu$_B$ in COX1. The Cu$_A$ site includes two Cu atoms of mixed valence and localizes to a hydrophilic COX2 domain in the intermembrane space (IMS). The Cu$_B$ site contains a single Cu ion that coordinates with a high-spin heme $a_3$ and a cross-linked His-Tyr cofactor to form the O$_2$ reduction, or 'active', site[5]. Near the active site, COX1 also harbors a low-spin heme $a$. During catalysis, electrons transfer from reduced cytochrome $c$ to the Cu$_A$ center in COX2. The electrons then flow from the Cu$_A$ site to heme $a$ in COX1 and are transferred to the Cu$_B$-heme $a_3$ center, which binds and reduces O$_2$ to H$_2$O [2]. The process is coupled to proton flow across the inner membrane to contribute to the proton gradient required for ATP synthesis by oxidative phosphorylation[6]. CcO deficiency is among the most recurrent causes of human mitochondrial disorders, frequently owing to mutations that impair CcO biogenesis and cause devastating cardio- and encephalomyopathies[7,8].

Mitochondrial CcO biogenesis is a modular process hypothesized to involve the independent maturation of COX1 and COX2 followed by sequential interactions with accessory subunits and assembly factors. The overall process requires over 30 nucleus-encoded assembly factors (Supplementary Table 1)[2], five of which are required to assemble the four-subunit $aa_3$-type CcO of $\alpha$-proteobacteria, such as *Rhodobacter sphaeroides*. They function to synthesize (COX10 and COX15) or deliver (SURF) heme $a$, and assemble Cu$_A$ (SCO) or Cu$_B$ (COX11) sites[2,9]. In mitochondria, heme $a$ and Cu insertion into COX1 and COX2 is essential for modular assembly[2]. In the current model, the soluble IMS Cu-chaperone COX17, which has a twin CX$_9$C structural motif and a CHCH Cu(I) binding motif, transfers Cu(I) to the IMS-exposed Cu-binding domains of IM-anchored COX11 and two SCO proteins (SCO1 and SCO2). COX11 and SCO1/SCO2 specifically and directly transfer Cu to the Cu$_B$ and Cu$_A$ sites, respectively. The human Cu$_A$ assembly mechanism is relatively well understood. It occurs in a module that contains the COX2 folding chaperone COX20, SCO1 for Cu transfer, SCO2 in Cu binding and disulfide reduction, and the IMS twin CX$_9$C protein COA6 cooperating as a copper chaperone or a thiol oxidoreductase[2,10–15]. Formation of the Cu$_B$ site in COX1 is less understood. The requirement of COX11 is only supported by studies in bacteria and yeast[9,16,17] and is assumed to occur similarly in human mitochondria. In yeast, COX11 activity is essential for Cu$_B$ assembly and depends on the twin CX$_9$C protein COX19, which prevents COX11 overoxidation[18]. Newly synthesized human COX1 associates with its folding chaperones COA3 and COX14 and another IMS twin CX$_9$C protein, CMC1[19]. Cu

insertion into COX1 was proposed to occur in this module, but interaction with human COX11 or COX19 has not been demonstrated.

Despite the wealth of biochemical and structural information available on human CcO biogenesis, the process of Cu$_B$ site assembly lacks understanding. Whether mechanisms are in place to couple heme $a$ biosynthesis and Cu$_B$ assembly, synchronize Cu delivery to Cu$_B$ and Cu$_A$, and coordinate the merging of the COX1 and COX2 assembly modules remains unknown. Herein, we set out to unravel these mechanisms by characterizing human cell lines lacking *COX11* and *COX19* physiologically and biochemically. We conclude that dynamic metallochaperone modules involving heme $a$ biosynthetic enzymes, COX1- and COX2-specific copper chaperones, which are regulated by several twin CX$_9$C proteins, including COX17, COX19, PET191, and COA6, coordinate the assembly of catalytic metal redox centers in human CcO. The regulatory mechanism is fundamental for CcO assembly and prevents the accumulation of heme-containing CcO assembly intermediates lacking copper, which induce oxidative-stress mediated cytotoxicity.

## Results

**COX19 is essential for human CcO biogenesis, but the absence of COX11 allows some residual CcO assembly and function.** We used the CRISPR-Cas9 technology to create stable human HEK293T cell line KOs for *COX11*, or *COX19* (Supplementary Fig. 1A–D, G). Compared to wild-type (WT), the *COX11*-KO line retained 50% of COX1 and 15% of COX2 steady-state levels and 15% of residual fully assembled CcO (labeled as CIV in the BN-PAGE assays) (Fig. 1A–C) and CcO activity (Fig. 1D), which supported 60% of respiratory capacity (Fig. 1E). By contrast, the COX1 levels in the *COX19*-KO line were 25% of WT, and COX2 was undetectable, resulting in the loss of holo-CIV, CcO activity, and cellular respiration (Fig. 1B, E and Supplementary Fig. 2A, C, D, G). The two cell lines accrue COX1-containing CIV sub-assemblies (Fig. 1C and Supplementary Fig. 2C, D), reflecting an early CcO assembly defect. These CIV assembly intermediates (labeled as S1-S4) contain COX1 but not COX2 (Fig. 1B and Supplementary Fig. 2C, D), and are similar to the intermediates that accumulate in cell lines carrying mutations that prevent COX2 incorporation (SURF1[20–23]) or metalation (SCO1[20,23]). The *COX11*-KO total mitochondrial cytochrome spectra showed an $\alpha$ peak corresponding to hemes $a + a_3$ of ~75% of WT (Fig. 1F), markedly higher than expected for the residual CcO activity in this line.

Approximately 15% of hemes $a + a_3$, compared to WT, was detected in the *COX19*-KO line despite lacking CcO. In both mutants, the spectra revealed a 5–6 nm blue shift in the $\alpha$ peak (597 nm *versus* 603 nm for WT) (Fig. 1F) indicating alterations in the environments surrounding the heme centers in COX1, as observed in *cox11$\Delta$ R. sphaeroides* strains[9]. It also implies that heme $a$ biosynthesis and delivery to COX1 can proceed independently of COX11 and COX19. All the phenotypes in the KO lines were restored to WT levels by reconstitution with the corresponding FLAG-tagged WT proteins (Fig. 1A, C and Supplementary Fig. 2A, C), which excluded CRISPR off-target effects. Whereas COX19 is essential for human CcO biogenesis, the absence of COX11 allows some residual CcO assembly— suggesting a potential alternative pathway for Cu$_B$ metalation in metazoans. This was unexpected given the essentiality of yeast Cox11 for CcO assembly. Heterologous complementation studies showed that the yeast and human COX11 proteins are not interchangeable, despite being stable in the heterologous mitochondrial compartments (Supplementary Fig. 3A–C). Culture media supplementation with CuCl$_2$, His-Cu(II), or

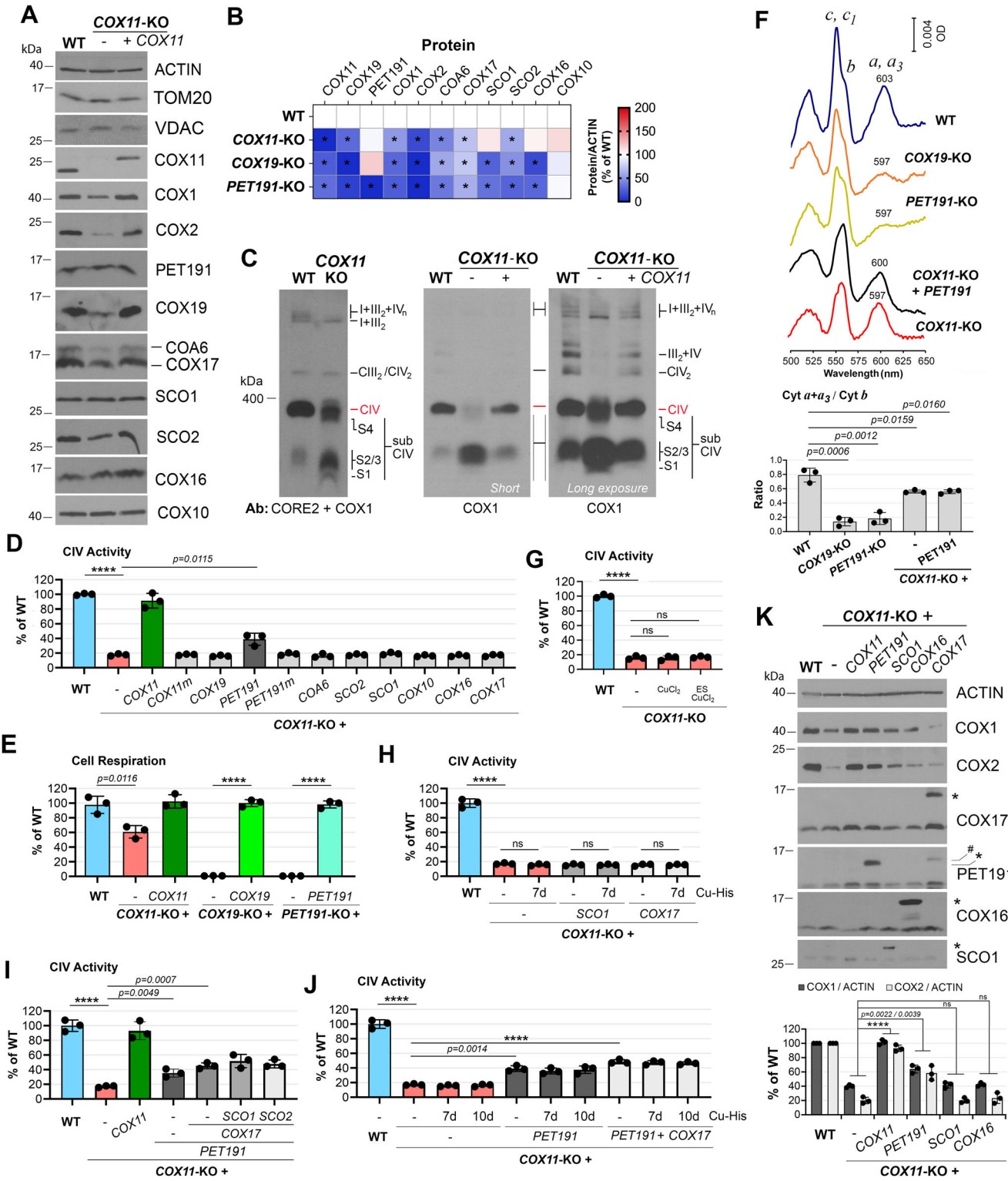

elesclomol-Cu(II), known to alleviate the CcO deficiency in *SCO2* patient cell lines and a *coa6Δ* yeast mutant[24,25], did not enhance CcO activity in *COX11*-KO cells (Fig. 1G, H, J). Thus, Cu$_B$ cannot self-assemble, and high Cu concentrations do not drive an alternative Cu-chaperone to substitute for COX11.

**Overexpression of *PET191* partially mitigates the CcO deficiency of *COX11*-KO cells.** COX19 contains four structural cysteine residues in two disulfides but lacks any Cu-binding

motif. It does not play any overlapping role with COX11, as overexpression of COX19 in the *COX11*-KO cell line did not improve its residual CcO activity or vice versa. (Fig. 1D and Supplementary Fig. 2G). Also, although SCO proteins have been implicated in the assembly of Cu$_B$ in α-proteobacteria with *cbb$_3$*-type oxidases[17], SCO1 or SCO2 overexpression did not restore CcO activity in *COX11*-KO cells even in the presence of exogenous Cu (Fig. 1D, H, I). Neither COA6 nor COX17 had a suppressive effect in *COX11*-KO cells (Fig. 1D, H). By contrast, our screen identified another IMS-resident twin CX$_9$C protein,

**Fig. 1 COX11 is dispensable for CcO biogenesis in human mitochondria. A** Steady-state levels of CcO subunits and assembly factors in *COX11*-KO cells. **B** Heat map showing the levels of CcO subunits and assembly factors normalized by ACTIN levels in *COX11*-KO, *COX19*-KO, and *PET191*-KO cells, presented as % of WT The map represents the average of three independent experiments. Two-sided unpaired *t*-test, *$p < 0.05$. **C** BN-PAGE analysis of ETC complexes (CIII$_2$, CIV, CIV$_2$) and supercomplexes (III$_2$ + CIV and I + III$_2$ + CIV$_n$) in *COX11*-KO cells reconstituted or not with WT *COX11*. CIV subassemblies (sub CIV) are labeled as S1-S3. S1 contains COX1-COX14-COA3-CMC1 and S2-S3 contain COX1-COX14-COA3-COX4-COX5a with or without assembly factors such as SURF1 or MITRAC7[19]. S4 is a CIV subcomplex formed by the off-pathway joining of the COX1 and COX3 assembly modules when COX2 is absent or in limited amounts[76]. **D, G–J** CIV (CcO) activity measured spectrophotometrically in the indicated cell lines. In **G, H, J**, the cells were incubated in the presence or absence of 1 mM CuCl$_2$ or 1 nM elesclomol (ES) + 1 mM CuCl$_2$ for 7 days. The bar graphs represent the average ± SD of three independent experiments. Black dots represent individual data points. Two-sided unpaired *t*-test, ****$p < 0.0001$. **E** Endogenous cell respiration measured polarographically in *COX11*-KO, *COX19*-KO, or *PET191*-KO cells reconstituted or not with the corresponding WT gene. The bar graphs represent the average ± SD of three independent experiments. Black dots represent individual data points. Two-sided unpaired *t*-test, ****$p < 0.0001$. **F** Total mitochondrial cytochrome spectra. Cytochromes *c* and *c$_1$* peak at 550 nm and cytochrome *b* peaks at 560 nm. The precise wavelength at which the absorbance of *a* + *a$_3$* cytochromes peak is annotated. The height of the peaks was calculated using the Quant mode of the UV-Probe software (Shimadzu) and expressed as the *a* + *a$_3$*/b ratio. In the lower graph, the data from three independent experiments were plotted with bars representing the mean ± SD. Black dots represent individual data points. Two-sided unpaired *t*-test. **G–J** CcO activity measured spectrophotometrically in the indicated cell lines. In **G, J**, the cells were incubated in the presence or absence of 1.5 mM Cu-His for 7–10 days with addition or not of elesclomol (ES) as indicated in **G**. The bar graphs represent the average ± SD of three independent experiments. Black dots represent individual data points. Two-sided unpaired *t*-test, ****$p < 0.0001$. ns, no significant. **K** Steady-state levels of CcO subunits and assembly factors in *COX11*-KO cells overexpressing CcO assembly factors. The bar graphs represent the average ± SD of three independent experiments. Black dots represent individual data points. Two-sided unpaired *t*-test, ****$p < 0.0001$. ns, no significant. Figures **A, C, K** are representative of three independent repetitions with similar results. Source data for **A, B,** and **D–K,** are provided as a Source Data file.

PET191 (*alias* COA5), whose overexpression enhanced CcO levels and activity in *COX11*-KO cells from 15% to 40% of WT, and up to 50% when co-overexpressed with COX17 (Fig. 1D, I, J). Additional overexpression of SCO1/ SCO2 (Fig. 1I) or exogenous Cu (Fig. 1J) had no additive effect on CcO activity. In addition to enhancing CcO activity, PET191 overexpression also elevated the steady-state levels of CcO catalytic subunits COX1 and COX2 (Fig. 1K), albeit the native CcO enzyme levels were only increased in the supercomplex structures (Supplementary Fig. 4F–G). PET191 overexpression slightly, yet significantly, modified the *a* + *a$_3$* cytochrome spectra (Fig. 1F and Source Data), with a fraction of the α peak red-shifted (~600 nm vs. 597 nm in *COX11*-KO) (Fig. 1F), suggesting that the binuclear center may be in or closer to its native state in this fraction.

PET191 is essential for CcO assembly in yeast[26] and weakly suppresses the growth defect on synthetic medium containing galactose of a *cox11Δ* strain (Supplementary Fig. 5A–C). Human *PET191* mutations lead to CcO deficiency and fatal hypertrophic cardiomyopathy[27]. We used transcription activator-like effector nucleases (TALENs) to generate HEK293T *PET191*-KO cell lines (Supplementary Fig. 1E–G), which had a phenotype akin to *COX19*-KO cells, with undetectable levels of COX2 and holo-CIV (Supplementary Fig. 2B, E). Thus, PET191 is essential for CcO biogenesis in human cells. PET191 has four structural cysteines forming two disulfides and two additional cysteines (C30 and C41), one of which is functionally essential in yeast[26]. Reconstitution of *PET191*-KO cells with WT *PET191* restored all the phenotypes while the mutant *PET191(C30A, C41A)* partially restored CcO activity (Supplementary Fig. 4A–C). The extra cysteines in PET191 could play a role in Cu or redox transfer since their mutation abolished PET191 capacity to suppress the *COX11*-KO CcO deficiency (Fig. 1D and Supplementary Fig. 4E–G).

**COX19, PET191, COA6, and COX2 are essential to maintain the redox state and copper-binding capacity of COX11.** Anchored in the inner membrane by a single transmembrane domain, mitochondrial COX11 has a C-terminal soluble head-group located in the IMS. The headgroup has a β-sheet immunoglobulin-like fold that contains the C217 and C219 residues in the human CcO, forming the conserved copper-binding CFCF motif[28] (Fig. 2A). A third cysteine residue, C121 in human CcO, locates immediately above the IMS surface of the membrane and

participates in copper transfer to COX1[16,17,29]. Functional bacterial COX11 is a dimer[28], with 2Cu(I) −4Cys centers facing the membrane and oriented to be near one of the two C121 residues while also interacting with two of the three COX1 histidine ligands of Cu$_B$, enabling Cu transfer[17].

We used an IAA/AMS-based reverse thiol-trapping approach to assess the native redox state of human COX11. Native COX11 can potentially be in three combinations of conformations and redox states: (i) *Dimeric Cu-bound COX11*. Stable copper binding to COX11 is absolutely dependent upon dimerization[17]. Cu-bound dimeric COX11 (Cu(I)·COX11) has two cysteines, C217 and C219, engaged in Cu coordination, and a reduced cysteine, C121, per monomer, which in the reverse thiol-trapping assay would result in the addition of 2 AMS molecules -to C217 and C219 (Fig. 2A). (ii) *Oxidized apo-COX11*. Although an intramolecular disulfide between C217 and C219 has not been observed in bacteria[17], it has been suggested in yeast[18]. In such a case, each monomer would bind 2 AMS molecules in the thiol-trapping assay (Fig. 2A). (iii) *Reduced apo-COX11*. In our assay, this scenario would result in no AMS binding (Fig. 2A).

Our reverse thiol-trapping analyses showed that in WT HEK293T cells, 50% of COX11 is Cu(I)·COX11 or oxidized monomeric COX11 (2AMS bound), but in *COX19*-KO and *PET191*-KO cells, COX11 is fully reduced in the apo-state (0 AMS bound) (Fig. 2B). The fraction of 2AMS-bound COX11 in *COX16*-KO was slightly increased compared to WT (Fig. 2C). By contrast, only apo-COX11 was detected in *COA6*-KO, which links COX11 to proteins, such as COA6, involved in COX2 metalation. We then assessed the COX11 redox state in WT 143B cells, in which more than 80% is Cu(I)·COX11, and homoplasmic cybrid cell lines lacking COX1, COX2, or COX3. The proportion of Cu(I)·COX11 + oxidized COX11 in *COX3* cybrids was similar to WT, 50% in COX1 cybrids, and ~5% in COX2 cybrids (Fig. 2D). These data functionally link COX11—as well as COX19 and PET191—to COX2 biogenesis.

To further analyze the dimerization state of COX11, we expressed functional COX11-FLAG in cell lines KO for *COX19*, *COX16* or *COA6*, which also contain endogenous COX11, and performed FLAG-immunoprecipitation (IP) assays. In the absence of either COX16 or COA6, a fraction of endogenous COX11 was co-immunoprecipitated (Supplementary Fig. 6A, B), indicating that human COX11 dimer formation is not strictly dependent of

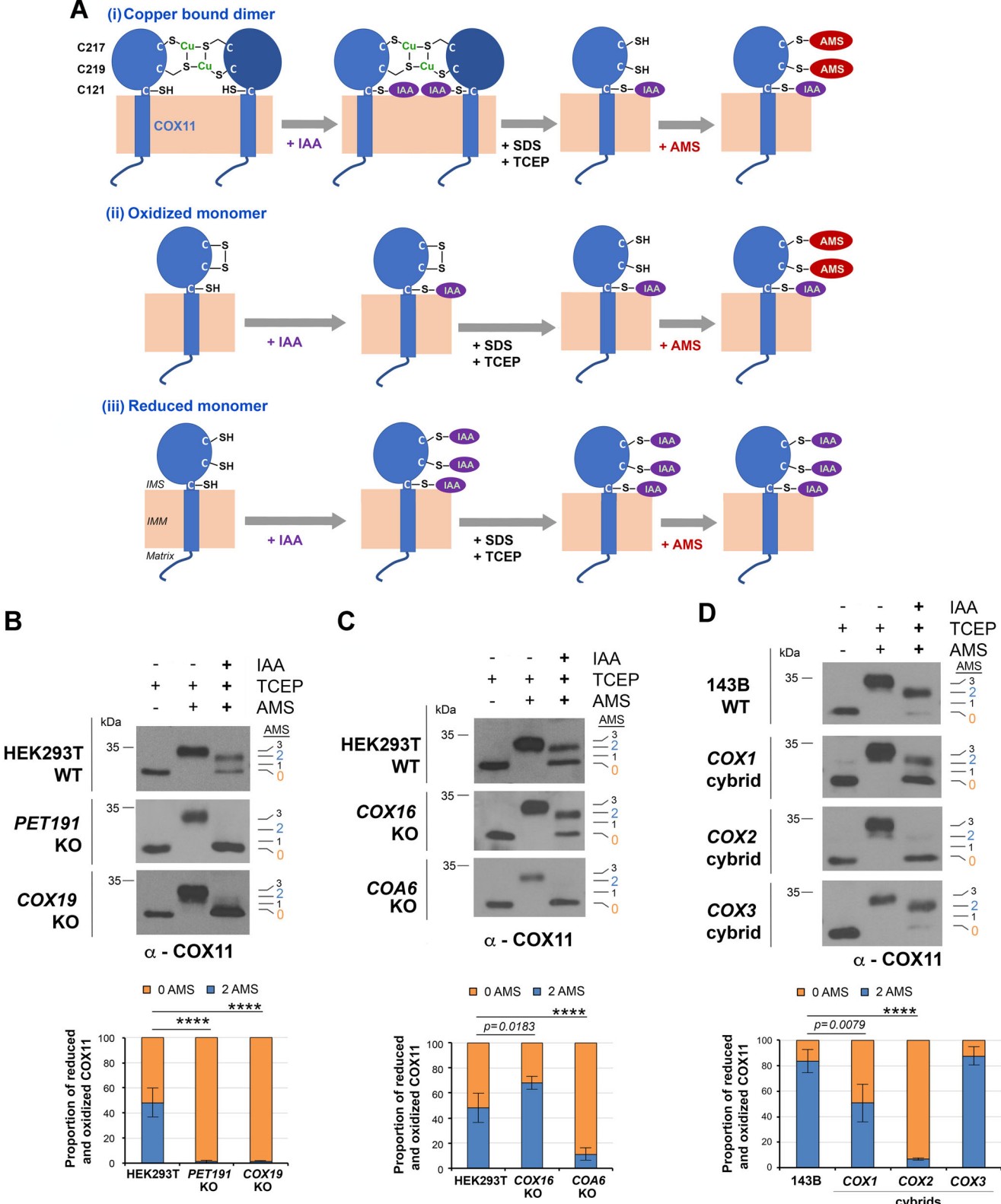

**Fig. 2 COX11 redox state and copper binding are regulated by PET191 and COX19. A** Reverse thiol trapping approach to detect the native cysteine residues inaccessible to a cell-permeable alkylating compound (2-iodoacetamide, IAA) that upon denaturation with SDS (sodium dodecyl sulfate) and full reduction with TCEP (tris(2-carboxyethyl)phosphine) are bound to AMS (4-acetamido-4'-maleimidylstilbene-2,2'- disulfonic acid), which adds ~540 Da per thiol. Three potential scenarios are depicted to account for copper (Cu)-bound COX11 dimer and oxidized or reduced COX11 monomers. **B–D** Reverse COX11 thiol trapping in the indicated cell lines. The bar graphs represent the average ± SD of four independent experiments. Two-sided unpaired $t$-test, ****$p < 0.0001$. Figures **B–D** are representative of three independent repetitions with similar results. Source data for **B–D** are provided as a Source Data file.

its redox and copper-binding states. No endogenous COX11 was co-immunoprecipitated with COX11-FLAG in the absence of COX19 (Supplementary Fig. 6A, B). Although the levels of endogenous COX11 are attenuated in COX19-KO cells, the data suggest that COX19 is necessary to support COX11 dimerization.

**COX11 physically interacts with newly synthesized COX2.** Metabolic labeling of mitochondrial polypeptides in whole cells with $^{35}$S-methionine followed by COX11 immunoprecipitation (IP) revealed that COX11 exhibits a stable interaction with newly synthesized COX2 but not with COX1 in WT HEK293T and 143B cells (Fig. 3A and Supplementary Fig. 7A).

The interaction persists in the absence of COX1 (Fig. 3A). However, it is abolished in the absence of COX20, a factor that stabilizes COX2 during insertion of its N-proximal transmembrane domain, or COX18, which transiently interacts with COX2 to promote translocation across the inner membrane of the COX2's C-tail that harbors the apo-Cu$_A$ site (Supplementary Fig. 7B). The lack of COX2 did not promote COX11-COX1 association (Supplementary Fig. 7B). Moreover, COX19 or PET191 did not stably interact with newly synthesized COX2 (Supplementary Fig. 7A, C). Instead, the three proteins—COX11, COX19, and PET191—affected the interaction between COX2 and known COX2 folding and Cu$_A$ metalation chaperones. The absence of COX11 diminished the interaction of COX2 with SCO2 by 30% and with COX20 or SCO1 by 80% (Supplementary Fig. 7D, G). Furthermore, SCO1, COA6, and COX16 failed to interact with the newly synthesized COX2 in the absence of PET191 (Supplementary Fig. 7E, G). The absence of COX19 produced similar effects, although 20% of the COA6-COX2 interaction was preserved. By contrast, the interactions of COX11 or SCO2 with COX2 were independent of PET191 or COX19 (Supplementary Fig. 7F–G). These data indicate that SCO2 and COX11 interact with apo-COX2 before the action of PET191 and COX19, and prior to the joining of SCO1, COA6, and COX16.

**COX1 and COX2 metallochaperones form stable and transient modules.** The steady-state levels of COX19 and COX11 were interdependent in all contexts tested. Both were depleted in *PET191*-KO cells, although COX19 was much more severely affected (Supplementary Fig. 2B). In contrast, the levels of PET191 were unaltered in the *COX19*-KO or *COX11*-KO lines (Fig. 1A, B and Supplementary Fig. 2A, C). COA6 and SCO2 levels were attenuated in *COX11*-KO cells to 50% of WT, and the levels of COA6, SCO2, SCO1, and COX16 were below 25% in the *COX19*-KO or *PET191*-KO lines, compared to WT (Fig. 1A, B and Supplementary Fig. 2A, B). Density gradient fractionation and Blue-Native (BN)-PAGE analyses of digitonin-extracted native protein complexes showed that COX11 is part of a ~250 kDa complex whose integrity depends on COX19 and PET191 (Supplementary Fig. 8A–D). The three proteins also form several complexes ranging from ~50 to 150 kDa (Supplementary Fig. 8A–D). Next, we performed IP assays in mitochondrial extracts from WT and each respective KO cell line reconstituted with a functional FLAG-tagged version of the corresponding protein, or candidate protein interactors relevant to COX1 and COX2 biogenesis—SCO1, SCO2, COX17, COA6, COX16, and COX10. The results (Fig. 3B, C and Source Data) indicate that COX11 stably interacts with COX2, the SCO1/SCO2 Cu$_A$ metallochaperones, the COX11 chaperone COX19, the heme $o$ synthase COX10, and the putative COX1 heme $a$ insertase SURF1. These interactions merge the COX1 and COX2 assembly modules. Moreover, COX17 forms stable interactions with SCO1/SCO2, and COX10, but not PET191 or COX11. Likewise, COA6 stably interacts with COX2, SCO1/SCO2, and COX16 (Fig. 3C and Source Data). To specifically test

whether COX19 binds to COX11 in a redox-sensitive manner, as shown in yeast[18], we use mitochondria from cell lines expressing only COX19-FLAG or COX11-FLAG, prepared extracts in the presence of 5 mM reduced glutathione (GSH), and performed reciprocal IP assays. The COX19-COX11 interaction remained stable (Supplementary Fig. 9).

To examine protein-protein interactions unbiasedly, we analyzed the IPs of COX11, COX19, and PET191 by LC-MS-MS. The data confirmed the interaction of these three proteins with SCO1, and the interactions of COX11 with COX19, SCO2, and COX2 (Fig. 3C). The MS studies also indicated potential interactions of COX11 and PET191 with the heme $a$ synthase COX15, the IMS CcO assembly factor COA7, and the holo-cytochrome $c$ synthase HCCS, as the most relevant proteins (Fig. 3C).

To capture transient protein-protein interactions with CcO metallochaperones occurring during CcO assembly, we used mitochondria purified from each respective line treated with the crosslinker DSP (dithiobis(succinimidyl propionate)) or the vehicle DMSO. Samples were extracted with 0.4% DDM (n-dodecyl β-D-maltoside), incubated with anti-FLAG agarose beads, and analyzed by immunoblotting to detect 13 candidate proteins: the 9 bites plus COX2, COX1, COA3, and SURF1. The original DSP data (see Source Data) is summarized in a compilation of heatmaps (Fig. 3D) that reflect the existence of several independent complexes with progressively overlapping compositions. For example, in WT mitochondria, COX11, SCO1, SCO2, and COX16 co-IP with all the 13 proteins analyzed; COX10, PET191, and COX17 do not interact with COX1, COX2, or SURF1; and COX11 and COX19 always co-assemble (Fig. 3D). PET191 does not co-IP with COX17 or COA6, suggesting it acts before their incorporation (Fig. 3D). Furthermore, we can determine the composition of CcO assembly modules by revealing the effect of the lack of each deleted protein on the stability and interaction of other factors from the same module (Fig. 3D). For example, the capacity of COX11 to interact with COX2 and its chaperones is hindered in the absence of COX19 or PET191; and PET191 coincides with COX17 in the same complex only in the absence of COA6 (Fig. 3D). Our data indicate that a complex containing presumably unmetalated Cu-chaperones and heme $a$ biosynthetic enzymes assembles and is stabilized by PET191, which dissociates from the complex before the association of COX17. COX17 delivers Cu to COX11 and SCO1/SCO2 before interacting, as a multi-metallochaperone module, with their target CcO subunits, first COX2 and then COX1. The COX1- and COX2-independent formation of the metallochaperone module is further supported by demonstrating that COX11 interactions—particularly with COX19, COX10, and the SCO1/SCO2 proteins—are preserved in WT cells treated with chloramphenicol for 72 h to inhibit mitochondrial protein synthesis, thus preventing any accumulation of COX1/2 (Fig. 3D and Supplementary Fig. 10C, D). The entrapment of COX10 in complexes with Cu$_A$ and Cu$_B$ chaperones until their Cu-loading suggests a concerted regulation of metalation of all the redox centers in CcO. Also, COA3 is detected in the same complexes that COX10. Both factors are undetectable in complexes containing the COX2-COX20-COX18 module until the COX1-COA3-COX14 module is incorporated. These data suggest a negative-feedback regulatory loop to coordinate COX1 synthesis and stabilization with heme $a$ and copper bioavailability.

**The redox and Cu-binding state of SCO1 and SCO2 are modified by Cu$_A$ and Cu$_B$ assembly chaperones.** Protein-protein interaction data predicts that the redox state and Cu binding to the Cu$_A$ metallochaperones SCO1 and SCO2 could depend on Cu$_B$ assembly chaperones. SCO1 and SCO2 have different roles in

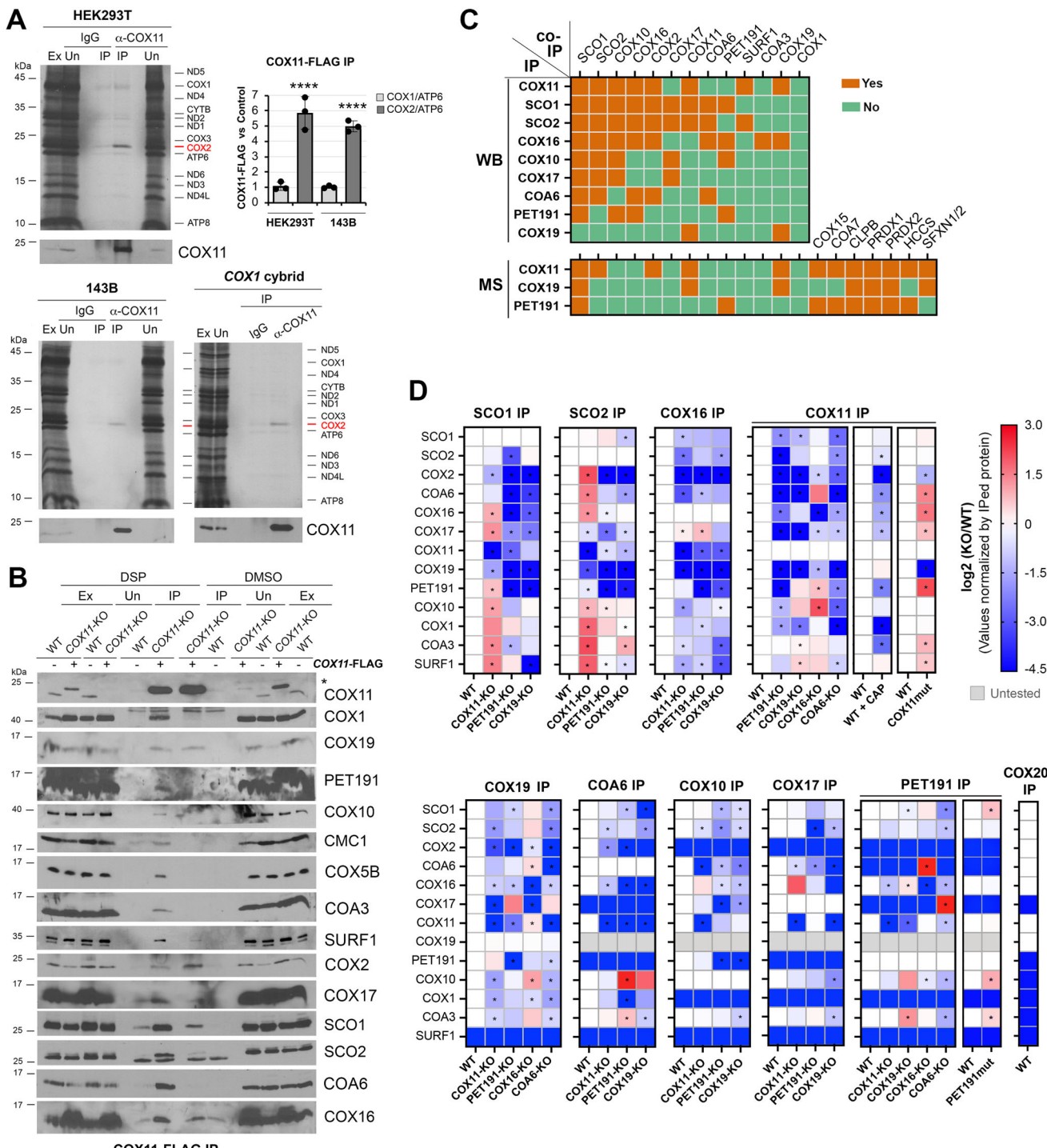

**Fig. 3 CcO subunits and metallochaperones undergo stable and transient interactions to drive and coordinate metal center assembly. A** Co-immunoprecipitation (co-IP) of endogenous COX11 with newly synthesized mitochondrial polypeptides, identified in the right-hand side. The graph shows the densitometry (average ± SD) of three independent experiments. Black dots represent individual data points. Two-sided unpaired *t*-test, ****$p < 0.0001$. The ratio "α-COX11 IP signal vs. control background signal" for two parameters, COX1/ATP6 and COX2/ATP6, is presented. **B** Co-IP of COX11-FLAG with CcO subunits and assembly factors in the presence or absence of the crosslinker DSP. Ex, extract; Un, unbound; IP, immunoprecipitate. **C** Stable interactions among CcO assembly factors determined by IP followed by either immunoblotting (WB) or mass spectrometry (MS). **D** Transient interactions among CcO assembly factors determined by IP in the presence of the crosslinker DSP (dithiobis(succinimidyl propionate)) or the vehicle DMSO (dimethyl sulfoxide) followed by immunoblotting. The heat maps show the average quantification of three independent experiments. Two-sided unpaired *t*-test. *$p < 0.05$. Figures **A**, **B** are representative of three independent repetitions with similar results. Source data for **A**, **B**, **D** are provided as a Source Data file.

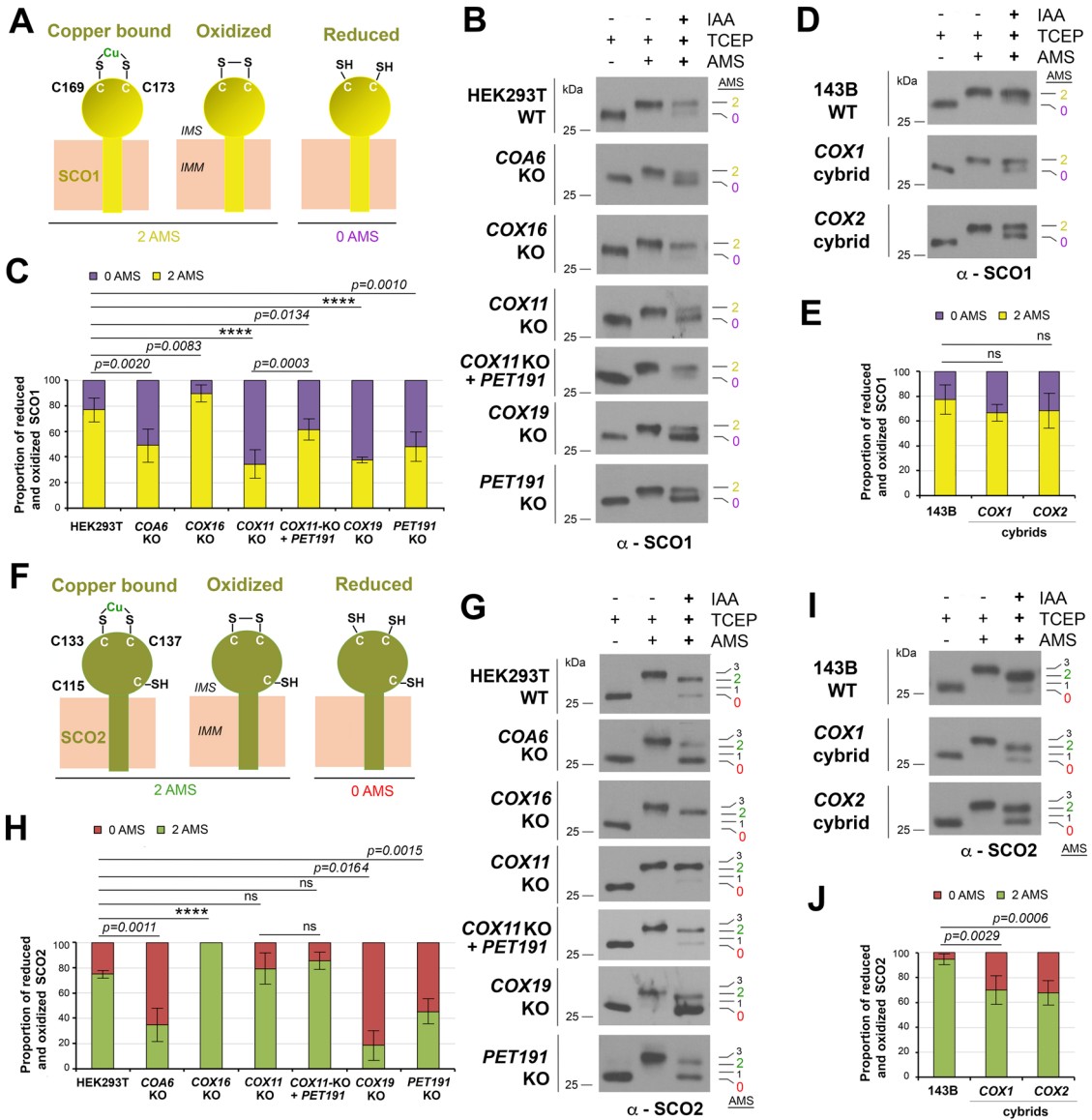

**Fig. 4 SCO1 and SCO2 redox state and copper binding are regulated by COX11, PET191, and COX19.** Potential redox and copper (Cu) binding states of SCO1 (**A**) or SCO2 (**F**). **B**, **D**, **G**, **I** Reverse SCO1 or SCO2 thiol trapping in the indicated cell lines. **C**, **E**, **H**, **J** The graphs show the quantification of six (for SCO1) or five (for SCO2) independent experiments as in **B**, **D**, **G**, **I**, respectively. The bars represent the mean ± SD. Two-sided unpaired t-test, ****$p < 0.0001$. ns, no significant. Figures **B**, **D**, **G**, **I** are representative of six (**B**, **D**) or five (**G**, **I**) independent repetitions with similar results. Source data for **B**–**D** and **H**–**J** are provided as a Source Data file.

$Cu_A$ assembly[11]. Cu(I)·COX17 delivers Cu(I) and electrons to oxidized SCO1, but only Cu(I) to SCO2, which may be reduced by COA6[2,14,15,30,31]. SCO1 selectively transfers Cu(I) ions to COX2 based on loop recognition, and SCO2 is a Cu-dependent thiol reductase of the $Cu_A$ cysteine ligands in COX2[32]. SCO1 and SCO2 have two essential cysteines (C169 and C173 in SCO1, and C133 and C137 in SCO2) arranged in a thioredoxin fold, which together with a conserved histidine constitute the Cu-binding residues. SCO2 has an additional cysteine residue, C115, which is not near the copper coordination center. Each protein can have the two key cysteines forming a disulfide, binding Cu, or reduced and Cu-free. In reverse thiol trapping assays, the first two scenarios would result in the addition of two AMS molecules and none in the latter scenario (Fig. 4A, F). For SCO1, 75% of the protein binds 2AMS in WT HEK293T (Fig. 4B, C) or 143B cells (Fig. 4D, E). The proportion was elevated to 90% in *COX16*-KO cells, decreased to 50% in *COA6*-KO or *PET191*-KO, and 35% in

*COX11*-KO or *COX19*-KO while it was restored to 60% in *COX11*-KO cells overexpressing PET191 (Fig. 4B, C). No changes were observed in COX1 or COX2 cybrids (Fig. 4D, E), suggesting processes independent of these core subunits. The data depicted a different pattern for SCO2. In WT HEK293T and *COX11*-KO cells, 75% of SCO2 bound 2 AMS but increased to 100% in *COX16*-KO cells (Fig. 4G, H). The absence of COA6 or PET191 lowered the proportion to 35–40%, and the absence of COX19 to 20% (Fig. 4G, H), suggesting SCO2-COX19 functional cooperation. In WT 143B cells, 90% of SCO2 bound 2 AMS, compared to 65% in the absence of COX1 or COX2 (Fig. 4I, J), which may reflect slower redox recycling of SCO2 in these scenarios.

**Mutant PET191(C30A, C41A) accumulate in CcO metallo-chaperone complexes that dictate the redox state of COX11, SCO1, and SCO2.** PET191 contains six cysteine residues and, as

predicted by AlphaFold[33,34], it is expected to fold in a helical hairpin analogous to COX17 and other twin CX$_9$C family proteins (Supplemental Fig. 11A). Four cysteines present in the structural motifs would form two disulfides. The other two, C30A and C41A, are found within the linker motif and could play roles beyond structural stabilization. A mutant variant PET191(C30A, C41A) is stable, partially functional (Supplementary Fig. 4A–C), and can interact with the same CcO assembly factors as WT PET191. When expressed in PET191-KO cells, either the WT or mutant protein interacts with COX11, COX19, COX10, SCO1, SCO2, COA3, and COX16 (Fig. 3D and Supplementary Fig. 4D). When expressed in COX11-KO cells, each still interacts with COX10, SCO1, SCO2, COA3, and COX16 (Supplementary Fig. 4D), suggesting the existence of an earlier metallochaperone complex involving these factors (Supplementary Fig. 11B). However, when expressed in COX11-KO cells, PET191(C30A, C41A) exerted a dominant-negative effect on CcO levels (Supplementary Fig. 4E–H). This could be due to the mutant protein partially compromising the mitochondrial import and lowering the organellar levels of the WT protein (Supplementary Fig. 11B) and competing with the WT variant for the interaction with other metallochaperones. Given the pronounced effects of loss of PET191 on the redox state of the cysteinyl sulfurs of COX11 (Fig. 2B), SCO1 (Fig. 4B), and SCO2 (Fig. 4G), all of which become more reduced, we tested the redox effects on these three factors of overexpressing mutant PET191. PET191(C30A, C41A) overexpression in HEK293T WT cells increased COX11 oxidation, while cells lacking PET191 increased oxidation of COX11, SCO1, and SCO2 to levels similar to WT (Supplementary Fig. 11C–E). The restoration of CcO assembly by PET191(C30A, C41A) is only partial, which could be accounted for by an increased resistance to be released from the apo-metallochaperone complex and substituted by COX17.

**Coordination of metal center assembly in COX1 minimizes the transient accumulation of pro-oxidant heme *a*-COX1 stalled intermediates**. The trapping of COX10 and COX15 in complexes with Cu$_A$ and Cu$_B$ chaperones, until the latter are Cu-charged, indicates a precise regulation of the metalation of all the redox centers in CcO to coordinate the process and avoid the accumulation of pro-oxidant intermediates. In yeast and human cells, endogenous superoxide production, measured with the fluorescent probes DHE and MitoSOX, positively correlates with CcO activity and cellular respiration, as observed for the COX11-KO, COX19-KO, and PET191-KO cells (Fig. 5A). However, the levels of H$_2$O$_2$ detected by the probe H$_2$DCFDA signal were enhanced in COX19-KO and PET191-KO cells (Fig. 5A). Heme *a*-Cox1-stalled intermediates are potentially cytotoxic to cells as shown in yeast by the H$_2$O$_2$-sensitivity of *sco1*Δ and *cox11*Δ mutants[3,35] and, to a lesser extent, in *pet191*Δ cells (Supplementary Fig. 5D). The *cox11*Δ mutation yields an H$_2$O$_2$-sensitivity phenotype dominant over other CcO mutations, including *pet191*Δ, which can be prevented by blocking heme *a* synthesis by deleting the PET117 assembly factor (Supplementary Fig. 5D). Similarly, the human COX19-KO and PET191-KO lines are hypersensitive to H$_2$O$_2$, a phenotype that can be ablated by COX10 silencing (Fig. 5B). These data indicate that the H$_2$O$_2$ sensitivity of COX19-KO and PET191-KO lines arises from pro-oxidant Cox1 assembly intermediates involving COX1 and heme *a*, which underscores the physiological need for coordination of heme *a* and Cu centers assembly during CcO biogenesis.

Mechanistically, the enhanced radical production measured in the absence of COX19 or PET191 could change the local redox environment and affect the redox state of the copper metallochaperones COX11, SCO1, or SCO2. However, this is unlikely

because the metallochaperones become more reduced or do not bind copper in COX19-KO or PET191-KO cells (0 AMS bound in the reverse thiol trapping experiments) (Figs. 2B and 4). Silencing of COX10 in HEK293T cells (Supplementary Fig. 12A, B) enhances levels of reduced- or apo- forms of COX1 and SCO1 (Supplementary Fig. 12C–E) similar to the absence of COX19 or PET191 (Figs. 2 and 4). Furthermore, COX10 silencing decreases the steady-state levels of PET191 and COX19 (Supplementary Fig. 12A). Altogether, these data along with our IP data (Fig. 3) indicate that COX10 is necessary for the formation of the early metallochaperone complexes required to promote the oxidation or copper metalation of COX11 and SCO1 (Supplementary Fig. 10B). Consistently, the absence of PET191 did not change the redox state of COX11 and SCO1 following COX10 silencing (Supplementary Fig. 12C–E). The redox pattern for SCO2 remained unaltered upon COX10 silencing in HEK293T WT cells and the mutant backgrounds tested (Supplementary Fig. 12E), which could indicate that SCO2 enters the assembly pathway downstream of COX10 release from early metallochaperone complexes or that its redox state is independent of the formation of these complexes.

**Copper binding to SCO1 and SCO2 depends on COX19 or copper metalation of COX11**. To probe the interaction of apo-COX11 with the other metallochaperones, we generated cell lines expressing non-functional but stable C217D and C219D variants of COX11 that mimic the SOH state of these residues (Supplementary Fig. 10E–H). Cysteines C217 and C219 form the conserved CFCF motif involved in copper binding[28]. The point mutations prevented COX19 accumulation and decreased COX2 levels as in COX11-KO cells, although the residual amounts of COX2 remained associated with COX11 (Fig. 3D and Supplementary Fig. 10H). Moreover, the FLAG-tagged C219D variant co-IPed SCO1, SCO2, COX10, and COX1, as did the WT COX11 protein. However, it did co-IP 2-3-fold more of COA6, COX16, COX17, PET191, COA3, and particularly SURF1, with which it forms a stable interaction (Fig. 3D and Supplementary Fig. 10H). These data represent an accumulation of COX1 and COX2 metallochaperone modules stalled in assembly. To assess whether the metallochaperone complexes formed in the presence of WT or mutant COX11 contain copper, we extracted them from purified mitochondria in native conditions without using cross-linkers, FLAG-immunoprecipitated the complexes, and analyzed them by inductively coupled plasma MS(ICP-MS) elemental analysis (Fig. 5C). ICP-MS revealed the presence of significant amounts of Cu in these stable complexes when containing WT COX11, and the absence of detectable bound Cu in complexes associated with mutant COX11 (Fig. 5D). Thus, either the presence of mutant apo-COX11 or, most likely, the absence of COX19, impedes Cu binding by SCO1/SCO2, and probably also by COX17, which is present in the metallochaperone modules. Moreover, IP of PET191-FLAG expressed in PET191-KO cells did not yield any significant Cu signal (Fig. 5A). Since PET191 stably interacts with SCO1, COX16, and COX10 (Fig. 3B), these data indicate that neither SCO1 nor PET191 bind measurable copper amounts in these complexes. On the contrary, IP of functional PET191-FLAG, but not PET191(C30A, C41A)-FLAG, over-expressed in COX11-KO cells, yielded Cu levels 40% of the WT COX11-FLAG IP (Fig. 5A). Because PET191 never interacts with COX17 and does not form stable complexes with SCO1 or SCO2 in the absence of COX11 (see Source Data), the results demonstrate that PET191 binds Cu in the absence of COX11, but in the presence of COX11, PET191 either does not bind Cu or binds it loosely. Hence, the PET191-driven alternative pathway for Cu$_B$ metalation might only operate when the canonical COX11-driven

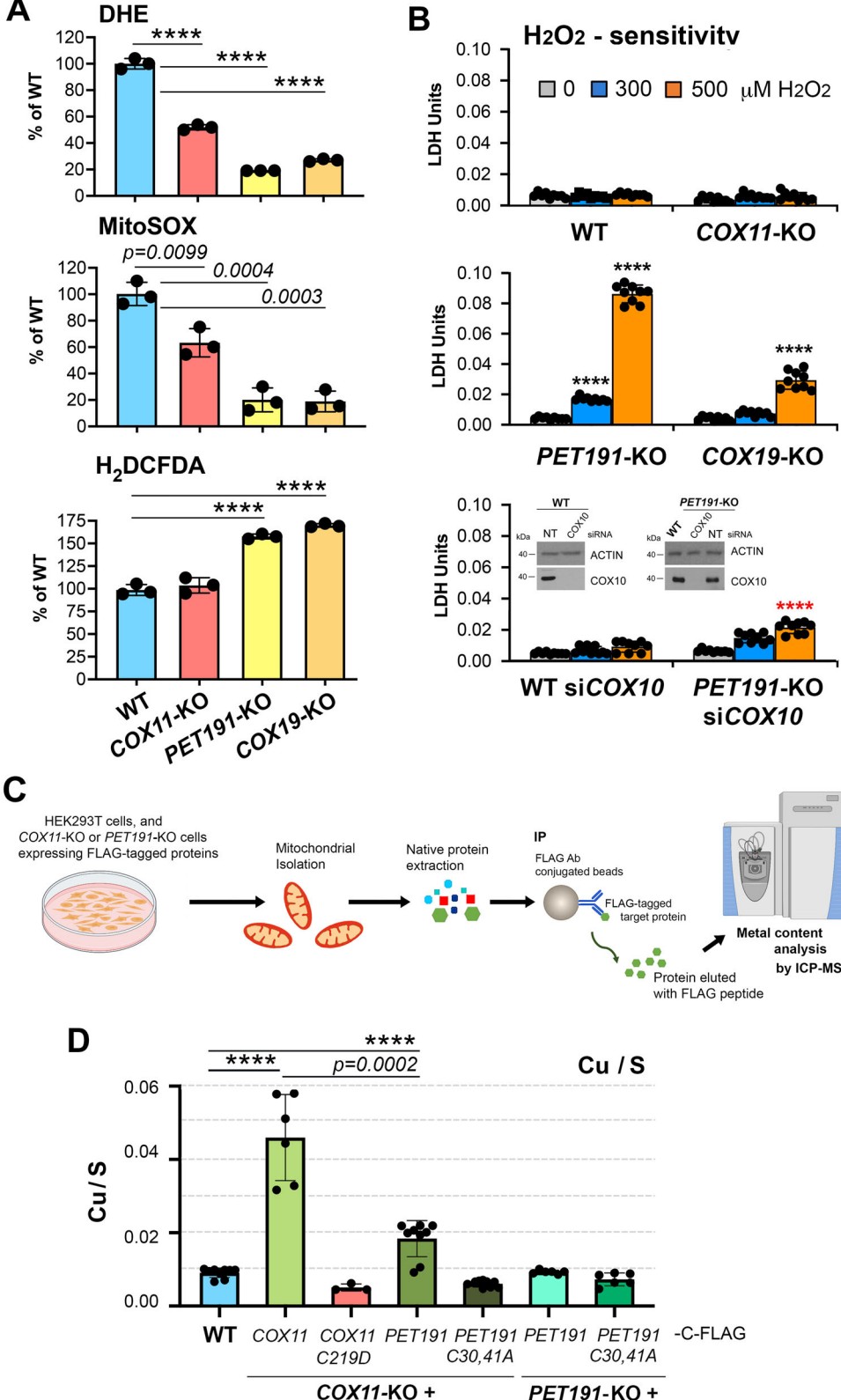

pathway is inefficient or overwhelmed, which could occur in response to stress or in a disease state.

## Discussion

This study highlights a set of dynamic protein-protein interactions among CcO-specific metallochaperones and assembly factors as well as regulatory checkpoints that ensure the coordinated and hierarchical assembly of metal centers in human CcO. By characterizing cell lines lacking COX11, COX19, or PET191 and the dynamic protein interactome of these proteins, we significantly advanced our understanding of the role these proteins play in CcO biogenesis, identified CcO metallochaperone complexes, and determined the order in which these proteins interact

**Fig. 5 COX11- and PET191-containing metallochaperone modules coordinate copper and heme center assembly to minimize accumulation of pro-oxidant deleterious CcO assembly intermediates. A** ROS generation in the indicated cell lines measured using superoxide-sensitive dyes dihydroethidium (DHE) and MitoSOX and the $H_2O_2$-sensitive dichlorodihydrofluorescein diacetate (CM-$H_2$DCFDA). The bar graphs are quantification (mean ± SD) of three independent experiments. Black dots represent individual data points. Two-sided unpaired $t$-test, ****$p < 0.0001$. **B** $H_2O_2$ sensitivity measured as the release of lactate dehydrogenase (LDH) to the growth medium. In the bottom graph, the inset shows immunoblot analysis of COX10 steady-state levels in WT and *PET191*-KO HEK293T cells following 10 days of treatment with siRNA-COX10 or non-targeting (NT) control. ACTIN was used as the loading control. The bar graphs are quantification (mean ± SD) of nine independent experiments. Black dots represent individual data points. Two-sided unpaired $t$-test, ****$p < 0.0001$. Black stars are for comparisons with WT, and red starts for comparison with *PET191*-KO. Immunoblots in this panel are representative of three independent repetitions with similar results. Source data for the immunoblots are provided as a Source Data file. **C** Experimental workflow for copper measurements. Mitochondria isolated from WT or KO cell lines expressing FLAG-tagged COX11 or PET191 were solubilized in native conditions (see details in the Methods section), then the extracts used for FLAG-immunoprecipitation (IP), the IPed proteins eluted, and analyzed their metal content by inductively coupled plasma MS (ICP-MS). The panel was created with Biorender.com. **D** Copper content in COX11- and PET191-containing metallochaperone modules measured by ICP-MS. Mitochondria were purified from *COX11*-KO or *PET191*-KO cells expressing the indicated FLAG-tagged proteins. The proteins were immunoprecipitated using anti-FLAG conjugated beads, and their metal content was assessed by ICP-MS. The data is expressed as total Cu normalized by [$^{34}$S]. The bar graphs are quantification (mean ± SD) of three (*COX11*-KO + *COX11-C219C*), six (*COX11*-KO + *COX11* and *PET191*-KO + *PET191* or + *PET191-C30A,C41A*) or nine (WT, COX11-KO + *PET191* or + *PET191-C30A,C41A*) independent samples. Black dots represent individual data points. Two-sided unpaired $t$-test, ****$p < 0.0001$. Source data for **A**, **B**, **D** are provided as a Source Data file.

with core subunits COX1 and COX2. Our research also provides a paradigm to understand how the synthesis, maturation, and assembly of CcO core subunits are concerted to prevent the accumulation of cytotoxic pro-oxidant intermediates in human cells.

This work discloses unanticipated levels of sophistication surrounding $Cu_B$ center assembly in human CcO. COX11 is essential for the assembly of $Cu_B$ in the $aa_3$-type CcOs of α-proteobacteria[9,17] and yeast mitochondria[16,36,37]. Our data indicate that COX11 is also the major $Cu_B$ insertion factor in human mitochondria but, contrary to what occurs in yeast, a COX11-independent route can contribute up to 15% of assembled and functional CcO in the absence of COX11. The evolutionary origin of COX11 has been linked to a need for specificity to distinguish the $Cu_B$ and $Cu_A$ sites[17]. In support of this model, many bacterial heme-Cu oxidases that do not have $Cu_A$ in addition to $Cu_B$ do not require COX11 for $Cu_B$ assembly, such as *R. sphaeroides* $cbb_3$-type CcO or quinol oxidases[9]. In these cases, the insertion of $Cu_B$ is catalyzed by SCO family proteins[17]. Unexpectedly, our screen for suppressors of the partial $Cu_B$ assembly defect of *COX11*-KO cells discarded SCO1 and SCO2 even in the presence of exogenous copper and identified the twin $CX_9C$ protein PET191/COA5 as a key contributor in human cells.

The data reported herein represent a significant step to unveil the mechanism of action for several elusive twin $CX_9C$ proteins in the mitochondrial IMS. The role of COX17 as a Cu-chaperone that transfers Cu to SCO1/SCO2 and COX11 is well established[30,38]. A physical interaction between COX17 and SCO1 was previously detected only in a high-throughput affinity purification-MS + Bio-ID study[39]. Including a crosslinker in our assays has allowed disclosing the COX17 interactome, which includes its downstream targets for copper transfer, COX11 and SCO1/SCO2, but not the CcO core subunits COX1 or COX2. Our work has identified PET191 as an apo-metallochaperone complex stabilizer that interacts with heme *a* biosynthetic enzymes and copper chaperones, most prominently with SCO1, and in doing so, we propose that it acts as the placeholder for COX17. Moreover, PET191, which contains two cysteines outside its structural twin $CX_9C$ motifs, can also promote $Cu_B$ assembly in the absence of COX11, a capacity abolished when the extra cysteines are mutated. We also report that COX19 plays a role in stabilizing COX11 and in facilitating COX2 metalation with SCO2. COX19 contains only the four cysteines in the twin $CX_9C$ motifs and, in yeast, it has been proposed to maintain the reduced state of this conserved third cysteine in Cox11 by engaging in a Cox19-Cox11 heterodimer[18], which was identified by deep

learning methods[40]. Several high-throughput affinity purification-MS[39,41] and deep learning-based structure modeling studies[42] anticipated the conservation of the COX19-COX11 complex in human mitochondria, which we have now demonstrated as part of an extensive network of interactions discussed below. The predicted structures of the yeast[40] and human[42] complexes are similar and show that COX19 does not make direct contact with any of the three cysteine residues in COX11. However, we show that a single point mutation in COX11 residues C217 or C219 does not affect the steady-state level of COX11 but renders the protein non-functional, which destabilizes its interaction with COX19.

Studies in yeast suggested that Cox11 could form an intramolecular disulfide between the cysteines in the CFCF motif that is dependent on Cox19 and requires the third Cox11 cysteine residue[18]. However, copper binding to COX11 was not taken into full consideration for data interpretation in this study[18]. Also, the COX11-COX19 interaction might have different properties in yeast and human mitochondria since our data indicate that the COX11-COX19 interaction is redox-insensitive while the Cox11-Cox19 interaction in yeast is redox-sensitive[18]. Furthermore, native monomeric COX11 does not form an intramolecular disulfide in bacteria and only dimerizes to coordinate two copper molecules intermolecularly[17]. We show that in human mitochondria, the oligomerization of COX11 does not strictly depend on the protein's redox or copper-binding states. COX11 is completely reduced in the absence of COX19 and several other factors, including PET191, COA6, or COX2, whose absence severely compromises the stability of COX19. Although we cannot discard the action of a currently unknown thiol reductase, we favor a model in which COX19 directly maintains COX11 in a conformation competent for homodimer copper coordination.

The yeast Cox11-Cox19 interaction is comparable to the interaction between the twin $Cx_9C$ protein Mdm35 and the lipid-binding protein Ups1[43,44], which suggests that twin $Cx_9C$ proteins could act as stabilizers or folding chaperones to control the function of specific IMS proteins[18]. The human COX11-COX19 and SCO1-PET191 interactions would support this hypothesis. The CcO assembly model drawn from our data (Fig. 6) proposes that PET191 is dissociated from SCO1 before it acquires copper from COX17, and COX19 is dissociated from COX11 before it delivers copper to COX1. Conformational changes in SCO1 and COX11 induced by the binding and release of their "specific" twin $CX_9C$ protein would regulate their functions, as proposed for the yeast proteins mentioned earlier[18]. Also, the twin $CX_9C$ proteins

## Coordination of metal center assembly in human mitochondrial CcO

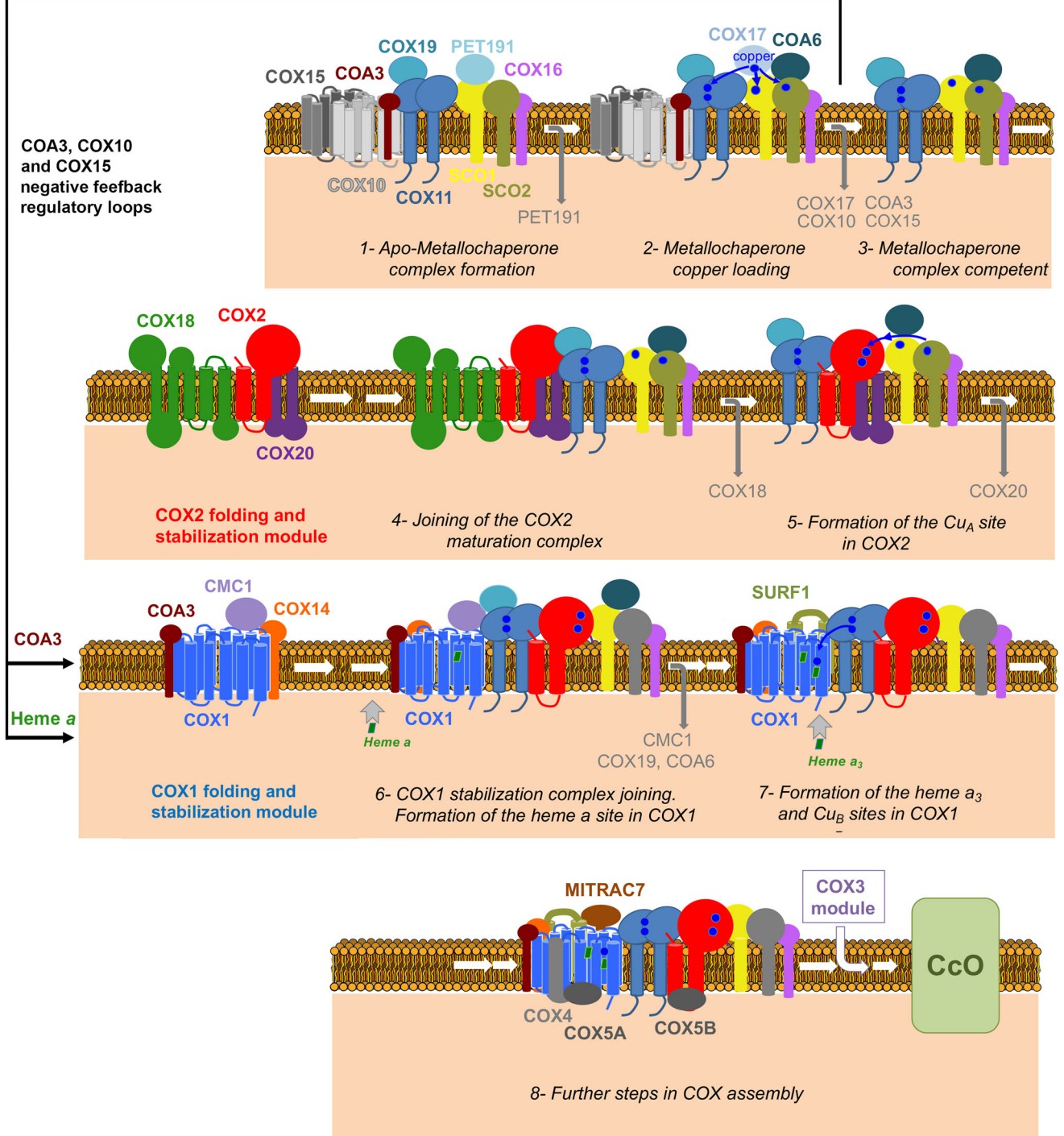

**Fig. 6 Model of CcO assembly depicting the coordination of metal center biogenesis.** The role of metallochaperone modules and the several regulatory checkpoints are indicated. See the text for details.

CMC1[19] and COA6[31] help stabilize their target proteins, COX1 and SCO2, respectively, making them competent to proceed or function in the CcO assembly process.

Our results support a concerted maturation of the COX1 and COX2 metal centers. COX11, COX19, and PET191 proteins have extensive stable and transient protein interactomes that include heme *a* biosynthetic enzymes and Cu$_A$ and Cu$_B$ metallochaperones. The data are compatible with recent high-throughput BioID studies in yeast that found Cox15 and Cox11 to localize to

mitoribosomal exit tunnel vicinity[45], and in human cells wherein, using SCO1 as a prey, researchers captured COX11, COX19, COX17, COX16, SCO2, COA6, COX5B-COX6B1, and COX15[39,46]. A high-throughput affinity purification-MS study also identified COX11-COX19-SCO1 physical interactions[39]. Furthermore, COX19, which partitions between the cytosol and the IMS in a copper-dependent manner, was proposed to contribute to transducing a SCO1-dependent mitochondrial redox signal that regulates cellular copper efflux[47].

As depicted in Fig. 6 and Supplementary Fig. 11C, our analyses demonstrate that PET191 stably interacts with the $Cu_A$ assembly factors SCO1-SCO2-COA6 and then recruits the COX11-COX19 pair prior to Cu acquisition. This dual Cu-center assembly module subsequently undergoes a substitution of PET191 by COX17, thereby enabling Cu transfer from COX17 to the downstream metallochaperones. In the mature CcO, most $Cu_A$ ligands lie within a region of COX2 that interacts strongly with a shallow depression on the surface of COX1, such that the accessibility of $Cu_A$ to other proteins is minimal. This implies that chaperone-mediated $Cu_A$ assembly must occur before COX2 associates with COX1. Concurrently, upon COX17 release, the metallochaperone module recruits COX2 and assembles $Cu_A$ before recruiting COX1 and building $Cu_B$. At this stage, COX1 must be hemylated at least in the $a$ site since it is detected in the COX19-KO and PET191-KO lines. Heme loading into the $a$ and $a_3$ sites may involve presently unknown heme delivery factors. However, studies in R. sphaeroides[9] and yeast cox11Δ strains[3] have shown that the heme $a_3$ moiety can be delivered to this site in the absence of Cu and that SURF1 plays a relevant role. Our data indicate that COX19 is released from the maturing CcO assembly intermediates once COX2 and COX1 have already been associated. This occurs before COX11 would transfer Cu to COX1, coinciding with the association of SURF1 to facilitate the assembly of the heme $a_3$ site. Hence, $Cu_B$ and the heme $a_3$ moiety could be incorporated simultaneously into the heterobimetallic center.

Several regulatory mechanisms have been proposed to facilitate the formation of the CcO metal centers while minimizing the formation of reactive assembly intermediates. Different translation regulation mechanisms coordinate COX1 synthesis and assembly in yeast[48] and human cells, wherein COA3 and COX14 interact with nascent COX1 polypeptides[49]. Heme $a$ biosynthesis and insertion into yeast Cox1 is regulated by assembly intermediates[50] harboring Cox15 and Shy1, the homolog of human SURF1[51]. Our data show that human COX11 forms complexes with COX10 and COX15, as well as COA3 in early apo-metallochaperone modules until they are Cu-loaded and competent to metalate COX2 and COX1. In that way, two previously undescribed negative-feedback regulatory loops coordinate COX1 synthesis, CcO metal center formation, and holo-complex assembly, thereby minimizing the accumulation of deleterious reactive CcO assembly intermediates.

The biomedical relevance of the COX11, COX19, and PET191 protein interactomes is highlighted by mutations in CcO subunits COX1 and COX2 and the assembly factors PET191, SCO1, SCO2, COA6, COA3, COX14, COX10, COX15, COX16, COX20, and SURF1, causing devastating human mitochondrial cardio- and encephalomyopathies. Therefore, this study may also shed light on the pathogenic mechanisms underlying these disorders.

## Methods

**Human cell lines and cell culture conditions**. Human HEK293T embryonic kidney cells (CRL-3216, RRID: CVCL-0063), HEK293 (CRL-1573, RRID: CVCL-0045), and 143B osteosarcoma cells (CRL-8303, RRID: CVCL-2270), were obtained from ATCC. Cybrid cell lines were constructed using enucleated control fibroblasts and the osteosarcoma 143B TK 206 rho zero cell lines[52]. COX1 and COX2 mutant cybrid cells carry a homoplasmic G6930A or G7896A mitochondrial mutation, respectively, that generates a stop codon and a truncated version of the protein[53,54]. The HEK293T COX16 knock-out (KO) cell line[55] was provided by Dr. Peter Rehling (University Medical Center Göttingen, Göttingen, Germany), and the COA6-KO cell line[15] by Dr. Michael Ryan (Monash University, Melbourne, Australia). The HEK293T COX18-KO and COX20-KO cell lines were previously reported by our group[12,56].

Cells were cultured in high-glucose Dulbecco's modified Eagle's medium (DMEM, Life Technologies) supplemented with 10% fetal bovine serum (FBS), 2 mM L-glutamine, 1 mM sodium pyruvate, 50 μg/ml uridine, and 1x GlutaMax (Thermo Fisher Scientific; Waltham, MA) at 37 °C under 5% $CO_2$. Cell lines were routinely tested for mycoplasma contamination.

For copper supplementation assays, the complete medium was supplemented with (i) 1.5 mM Cu-His, (ii) 1 mM $CuCl_2$, or (iii) 1 nM elesclomol (Selleckchem, STA-4783) + 1 mM $CuCl_2$.

**Generation of knockout cell lines and plasmid transfection**. To create stable human PET191-KO lines in HEK293T cells, two gene-specific pairs of TALEN constructs were obtained from Thermo-Invitrogen. The pair of left and right TALENs was designed to target the corresponding exon 1 (Supplementary Fig. 1). To create stable human COX19-KO and COX11-KO lines in HEK293T cells, we used CRISPR-CAS9 guide RNAs obtained from OriGene (KN210770 and KN203238, respectively) designed to bind the exon 1 region of the corresponding gene (Supplementary Fig. 1).

$1.5 \times 10^6$ HEK293T cells were transfected with 4 μg of the right and left TALEN plasmids as a pair with 1 μl of EndoFectin (GeneCopoeia; Rockville, MD) pre-incubated in 200 μl of OptiMEM-I (Thermo Scientific). Alternatively, HEK293T cells were transfected with 2 μg of donor DNA plasmid with puromycin resistance and gRNA with 1 μl of EndoFectin pre-incubated in 200 μl of OptiMEM-I. After 16 h of incubation, media were changed to complete culture medium. Four consecutive transfections were performed every 3 days. In the case of CRISPR-CAS9 transfection, the cells were selected with 2.5 μg/ml puromycin for 3 weeks. Next, the cells were sorted as single cells in 96-well plates. Single clones were grown and screened by immunoblotting against PET191, COX11, or COX19 antibodies (Supplementary Fig. 1), and by genotyping[12]. For genotyping, the PET191 locus was sequenced using the oligonucleotides PET191-Forward and PET191-Reverse, COX11-Forward and COX11-Reverse were used to sequence the COX11 locus and COX19-Forward and COX19-Reverse were used to sequence the COX19 locus (Supplementary Table 2 and Supplementary Fig. 1).

The KOs cell lines were reconstituted with C-terminal Myc-DDK-tagged wild-type version of the corresponding gene and other genes. The human genes PET191, COX19, COA6, COX17, COX16, and COX10 were obtained in plasmids from OriGene. The genes COX11, SCO1, and SCO2 were PCR-amplified from cDNA using primers listed in the Supplementary Table 2. The Saccharomyces cerevisiae COX11 gene was PCR-amplified from genomic DNA using primers listed in the Supplementary Table 2. All genes were cloned under the control of a truncated Δ5-CMV promoter[57] in the Δ5-pCMV6-A-Myc-DDK-Hygro plasmid (created from OriGene Technologies; Rockville, MD; PS100024) using Sgf1 and Mlu1 sites. To generate mutant variants of COX11 and PET191, we used the Q5® Site-Directed Mutagenesis Kit from NEB. ~10 pg of template Δ5-pCMV6-A-Myc-DDK-Hygro-COX11 or Δ5-pCMV6-A-Myc-DDK-Hygro-PET191 vector were used, along with the primers COX11_mut_C217A_F, COX11_mut_C217A_R, COX11_mut_C219A_F, COX11_mut_C219A_R, PET191_mut_C30A_F, PET191_mut_C30A_R, PET191_mut_C41A_F, PET191_mut_C41A_R (Supplementary Table 2), designed to include the codon to be mutated. After exponential amplification and the treatment with kinase and ligase, 2.5 μl of the reaction were transformed in competent Escherichia coli cells. For transfection of all constructs, we used 1 μl of EndoFectin mixed with 1–2 μg of vector DNA in OptiMEM-I media according to the manufacturer's instructions. Media was supplemented with 200 μg/ml of hygromycin after 24 h and drug selection was maintained for at least 1 month.

For co-overexpression of PET191 and COX17, COX11-KO cells were transfected with 1 μl of EndoFectin mixed with 1–2 μg of each vector DNA in OptiMEM-I media according to the manufacturer's instructions. The cells were selected in 200 μg/ml of hygromycin after 24 h. After 3 weeks of selection, the pool of cells was sorted as single cells in 96-well plates. Single clones were grown and screened by immunoblotting against PET191 and COX17 with respective antibodies and clones expressing both proteins were selected. For co-overexpression of PET191 and COX17, and one of the SCO genes, SCO1 and SCO2 were cloned into the pCMV6-A-BSD plasmid (OriGene Tech., PS100022) using Sgf1 and Mlu1 sites. The selected clones of co-overexpressing PET191 and COX17 were transfected with 1 μl of EndoFectin mixed with 1–2 μg of SCO1 or SCO2 expression constructs in OptiMEM-I media according to the manufacturer's instructions. The cells were selected in 50 μg/ml of blasticidin after 24 h and for 3 weeks in total.

**siRNA transfection**. HEK293T cells or PET191-KO were grown on 6-well plate at 10% confluency and transfected using 5 μl of Lipofectamine RNAiMAX (Thermo Fisher Scientific) mixed with either 10 nM of COX10 (GGUGCCAUUUGACUCA AACtt) siRNA (Ambion-Life Technologies; Austin, TX); or scrambled control [BLOCK-iT Alexa Fluor (Thermo Fisher Scientific)] in 200 μl OptiMEM-I (Thermo Scientific), according to the manufacturer's specifications. The cells were silenced for 10 days, and the silencing efficiency was confirmed by immunoblotting.

**Whole-Cell extracts and mitochondria isolation**. For SDS-PAGE electrophoresis, pelleted cells were solubilized in RIPA buffer (25 mM Tris-HCl pH 7.6, 150 mM NaCl, 1% NP-40, 1% sodium deoxycholate, and 0.1% SDS) with 1 mM PMSF (phenylmethylsulfonyl fluoride) and mammalian protease inhibitor cocktail (Sigma). Whole-cell extracts were cleared by 5 min centrifugation at $20,000 \times g$ at 4 °C.

Mitochondria-enriched fractions were isolated from at least ten 80% confluent 15-cm plates as described previously[12,58,59]. Briefly, the cells were resuspended in ice-cold T-K-Mg buffer (10 mM Tris-HCl, 10 mM KCl, 0.15 mM $MgCl_2$, pH 7.0) and disrupted with 10 strokes in a homogenizer (Kimble/Kontes, Vineland, NJ).

Using a 1 M sucrose solution, the homogenate was brought to a final concentration of 0.25 M sucrose, and a postnuclear supernatant was obtained by centrifugation of the samples twice for 5 min at $1000 \times g$. Mitochondria were pelleted by centrifugation for 20 min at $10,000 \times g$ and resuspended in 0.25 M sucrose, 20 mM Tris-HCl, 40 mM KCl, 10 mM MgCl$_2$, pH 7.4.

**Depletion of CcO core subunits.** To deplete the cells from mtDNA-encoded polypeptides, including CcO core subunits, COX1, COX2, and COX3, WT HEK293T and *COX11*-KO cells stably expressing COX11-FLAG were treated with 200 μg/ml chloramphenicol for 10 days to inhibit mitochondrial protein synthesis. As a control, cells were treated with the same volume of the vehicle (100% ethanol). The medium was changed every 48 h and supplemented with fresh chloramphenicol solution. Mitochondria were isolated as described above.

**Denaturing and native electrophoresis, followed by immunoblotting.** Protein concentration was measured by the Lowry method[60]. 40–80 μg of mitochondrial protein extract was separated by denaturing SDS-PAGE in the Laemmli buffer system[61]. Then, proteins were transferred to nitrocellulose membranes and probed with specific primary antibodies against the following proteins: COX11 (dilution 1:1000; OriGene Tech., TA323960), PET191 (dilution 1:500; Sigma; St. Louis, MO; HPA057768), COX19 (dilution 1:500; Sigma, HPA021226), β-ACTIN (dilution 1:2000; Proteintech; Rosemont, IL; 60008-1-Ig), COX1 (dilution 1:2000; Abcam; Cambridge, MA; ab14705), COX2 (dilution 1:1000; Abcam, ab110258), COA6 (dilution 1:500; Sigma, HPA028588), COX17 (dilution 1:500; OriGene Tech., TA315013), COX16 (dilution 1:1000; Proteintech, 19425-1-AP), COX10 (dilution 1:1000; Sigma, HPA032005), COA3 (dilution 1:1000; Sigma, HPA031966), SCO1 and SCO2 (PRAB4980 and PRAB4982 according to P. Rehling catalog; each at dilution 1:500; kind gift of P. Rehling who generated, validated, and cited them in PMID: 29381136[55]), CMC1 (dilution 1:1000; Sigma, HPA043333), COX5B (dilution 1:1000; Santa Cruz Biotech.; Dallas, TX; sc-374417), SURF1 (dilution 1:1000; Abcam, Ab155251), CORE2 (dilution 1:2000; Abcam, ab14745), FLAG (dilution 1:1000; Sigma, F3165), COX20 (dilution 1:1000; Sigma, HPA045490), HIGD2A (dilution 1:1000; Sigma, HPA042715), HA (dilution 1:1000; Thermo Fisher Scientific, 71-5500), Porin (dilution 1:1000; Abcam, ab110326). Horseradish peroxidase-conjugated anti-mouse or anti-rabbit IgGs were used as secondary antibodies (dilution 1:10,000; Rockland; Limerick, PA). β-ACTIN was used as a loading control. Signals were detected by chemiluminescence incubation and exposure to X-ray film. Optical densities of the immunoreactive bands were measured using the ImageLab (Biorad; Hercules, CA) software or the ImageJ software version 1.53r in digitalized images.

Blue-Native polyacrylamide gel electrophoresis (BN-PAGE) analysis of mitochondrial respiratory chain complexes, supercomplexes, and CIV assembly intermediates in native conditions was performed as described previously[62,63]. To extract mitochondrial proteins in native conditions, we pelleted and solubilized mitochondria in 100 μl buffer containing 1.5 M aminocaproic acid and 50 mM Bis-Tris (pH 7.0). After optimizing solubilization conditions, we used digitonin at the optimal 1:2 (digitonin: protein) proportion. Solubilized samples were incubated on ice for 15 min and pelleted at $20,000 \times g$ for 30 min at 4 °C. The supernatant was supplemented with 10 μl of sample buffer 10X (750 mM aminocaproic acid, 50 mM Bis-Tris, 0.5 mM EDTA (ethylenediaminetetraacetic acid), 5% Serva Blue G-250). Native PAGE™ Novex® 3–12% Bis-Tris Protein Gels (Thermo Fisher) gels were loaded with 60–100 μg of mitochondrial protein or total cell extracts. After electrophoresis, proteins were transferred to PVDF membranes using an eBlot L1 protein transfer system (GenScript, Piscataway, NJ) and used for immunoblotting.

**Characterization of the MRC and oxidative phosphorylation system.** The enzymatic activity of mitochondrial respiratory chain Complex IV (cytochrome *c* oxidase, CcO) was assessed spectrophotometrically using frozen-thawed cells by following the oxidation of exogenous reduced cytochrome *c* as reported[64]. Values were normalized by total protein concentration measured by the Lowry method[60].

Endogenous cell respiration was measured polarographically at 37 °C using a Clark-type electrode from Hansatech Instruments (Norfolk, United Kingdom). Cell respiration was assayed in cultured cells as reported[64]. Briefly, trypsinized cells were washed with respiration buffer (RB) containing 0.3 M mannitol, 10 mM KCl, 5 mM MgCl$_2$, 0.5 mM EDTA, 0.5 mM EGTA, 1 mg/ml BSA and 10 mM KH$_2$PO$_4$ (pH 7.4), and resuspended at $\sim 2 \times 10^6$ cells/ml in 0.5 ml of the same buffer, air-equilibrated at 37 °C. The cell suspension was immediately placed into the polarographic chamber to measure endogenous cell respiration. The specificity of the assay was determined by inhibition of complex IV activity and then respiration, with 0.8 μM KCN. Values were normalized by total cell number.

To obtain total mitochondrial cytochrome spectra, we extracted cytochromes from 8 mg of WT or respective KO mitochondria by resuspending them in 1190 μl of water with 70 mg of KCl (to 670 mM concentration), then adding a buffer containing 50 mM Tris, pH 7.5, and 1% potassium deoxycholate in a final volume to 1.4 ml, and gently mixing by pipetting. The extracts were cleared by centrifugation at $21,000 \times g$ for 20 min at 4 °C. Sodium cholate was added to the clear supernatant to a 1% final concentration. Half of the solution was reduced with sodium dithionite, while the other half was oxidized with potassium ferricyanide, and the difference spectra of the reduced versus oxidized extracts between 500 to

650 nm were recorded at room temperature using a UV-2401PC Shimadzu spectrophotometer. The wild-type α absorption bands corresponding to cytochromes *a* and *a$_3$* have maxima at 603 nm; the maxima for cytochrome *b* and for cytochrome *c* and *c$_1$* are 560 and 550 nm, respectively. The heights of the peaks and the area under the peaks in three independent experiments were calculated using the of the Quant mode of the UV-Probe software (Shimadzu) and expressed as the $a + a_3/b$ ratio. Similar values were obtained when using the peaks' height and the area under the peaks.

**Pulse labeling of mitochondrial translation products.** Mitochondrial protein synthesis was determined by pulse-labeling 70–80%-confluent 15 cm plates of HEK293T, KO cell lines, 143B, or cybrid cell lines. The cell cultures were incubated for 30 min in DMEM without methionine and then supplemented with 100 μl/ml emetine to inhibit cytoplasmic protein synthesis as described[12]. Cells were labeled for 30 min (Pulse) at 37 °C with 100 μCi/ml [$^{35}$S] methionine (PerkinElmer Life Sciences, Boston, MA). After incubation, the cells were washed once with 1X PBS, collected by trypsinization, and whole-cell extracts were prepared by solubilization in RIPA buffer (1% NP-40, 0.1% SDS, 0.5% Na-deoxycholate, 150 mM NaCl, 2 mM EDTA, and 50 mM Tris-HCl, pH 8.0) supplemented with 1 mM PMSF, and 1X EDTA-free mammalian protease inhibitor cocktail (Roche # 11836170001). After incubation, the cells were washed, trypsinized, collected, and proteins were extracted. 100 μg of each sample was separated by SDS-PAGE on a 17.5% polyacrylamide gel, transferred to a nitrocellulose membrane and exposed to Kodak X-OMAT X-ray film. The membranes were then probed with a primary antibody against β-ACTIN as a loading control.

For immunoprecipitation assays, pelleted cells were extracted in 800 μl of PBS, 0.4% n-dodecyl-β-D-maltoside (DDM), 1 mM PMSF, and 8 μl of protease inhibitor cocktail (Sigma, P8340) for 10 min on ice. After centrifugation at $20,000 \times g$ for 30 min at 4 °C, the extract (Ex) was incubated overnight with 25 μl of protein A agarose beads with rabbit IgG (negative control) or specific antibody raised in rabbit. The unbound material (Un) was removed, and the beads were washed five times in 1 ml PBS with 0.05% DDM. Then, the beads were incubated at 37 °C for 1 h in 60 μl of Laemmli buffer 2 x to release bound material (IP). Equivalent amounts of all fractions were separated by SDS-PAGE using a 17.5% polyacrylamide gel. Gels were transferred to a nitrocellulose membrane and exposed to X-ray film.

**Immunoprecipitation.** For immunoprecipitation of FLAG-tagged proteins, 1 mg of mitochondria of KOs with FLAG-tagged or wild-type HEK293T (as negative control) were incubated in 1 mM DSP (dithiobis[succinimidylpropionate], Thermo Scientific) or in DMSO carrier control for 1.5 h at 4 °C. DSP is thiol-cleavable, primary amine-reactive with an 8-carbon spacer arm crosslinker. The reaction was stopped by incubation with 10 mM Tris-HCl solution, pH ~7.4. After spin, the mitochondria were extracted in 600 μl of PBS, 0.4% DDM, 1 mM PMSF, and 8 μl of protease inhibitor cocktail (Sigma, P8340) for 10 min on ice. After centrifugation at $20,000 \times g$ for 30 min at 4 °C, the extract (Ex) was incubated for 4 h at 4 °C with 25 μl of beads conjugated with a FLAG antibody (anti-DYDDDDK beads, Sigma), previously washed in PBS. The unbound material (Un) was removed, and the beads were washed five times in 1 ml of PBS 0.05% DDM. Then, beads were incubated for 30 min in 65 °C with 100 μl of Laemmli buffer 2 x to release bound material (IP). Equivalent amounts of all fractions were analyzed by SDS-PAGE and immunoblotting.

To test the redox-sensitivity of the COX11-COX19 interaction, 1 mg of mitochondria were pre-incubated in the presence of 5 mM GSH or left untreated for 15 min at 4 °C. Following two washes with PBS, treated and non-treated mitochondria were used for extraction with 0.4% DDM and immunoprecipitation assays as described earlier.

**Immunoprecipitation and LC-MS-MS analysis.** For LC-MS-MS analysis, the samples were immunoprecipitated as described above. The bound material was extracted from the beads by incubation with a 62.5 mM Tris-HCl pH 6.5 and 2% SDS solution. The extracted proteins were precipitated using 100% ice-cold acetone overnight at −20 °C to remove residual detergent. The next day, the samples were centrifuged at $20,000 \times g$ for 15 min at 4 °C. The pellet was washed with 100% ice-cold acetone. The final pellet was air-dried and analyzed by LC-MS/MS using the services of the Keck Biotechnology Proteomics center at Yale University, and the SPARC BioCentre (Molecular Analysis) at The Hospital for Sick Children in Toronto (Canada). Data was visualized using the proteome software Scaffold v 5.

**Thiol trapping analyses of COX11, SCO1, and SCO2.** We used a reverse thiol trapping assay to analyze the *in organello* redox state of cysteines in COX11, SCO1, and SCO2 state of cysteines, as described previously[65]. Briefly, 100 μg of mitochondria purified from WT or KO cell lines were first incubated for 30 min under isotonic conditions with 80 mM of membrane-permeable iodoacetamide (IAA), a compound that covalently binds free thiols. After incubation, the samples were washed twice with 1 ml of PBS to remove unbound IAA. Mitochondria were then resuspended in 60 μl of PBS and solubilized in the presence of 0.2% SDS. Next, to reduce all the native oxidized cysteines (not bound by IAA), 5 mM of the reducing agent TCEP (tris(2-carboxyethyl)phosphine) was added to the samples that were

incubated for 12 min at 95 °C and let to cool down. The cysteines that IAA had not blocked were subsequently identified by incubation for 1 h at room temperature in the dark with 4.5 mM AMS (4-acetamido-4′-maleimidylstilbene-2,2′-disulfonic acid), a compound that binds free thiols, adding ~0.5 kDa for every molecule bound. Complete reduction of COX11, SCO1 or SCO2 was tested by reducing the proteins first with TCEP, and then adding excess IAA, followed by AMS addition. The maximum shift of reduced COX11, SCO1, or SCO2 was determined by reducing the sample with TCEP during extraction, quickly followed by treatment with AMS. Finally, the samples were supplemented with non-reducing Laemmli sample buffer and analyzed by non-reducing SDS-PAGE (12% for COX11 or 10% for SCO1 and SCO2) followed by immunoblotting.

**Sucrose gradient sedimentation analysis**. Sucrose gradient sedimentation analyses were performed as described previously[66]. One mg of WT or respective KO mitochondria were extracted in 300 μl of 20 mM Tris pH 7.5, 50 mM KCl, 1% digitonin, 1 mM PMSF, and 1 x protease inhibitor cocktail. The lysate was spun at $20,000 \times g$ for 30 min at 4 °C, and the clarified extract was loaded onto a 5 ml 7–20% sucrose gradient containing 20 mM Tris pH 7.5, 50 mM KCl, 0.1% digitonin, and 1 mM PMSF buffer and centrifuged at $150,000 \times g$ for 15 h at 4 °C. The gradients were fractionated into 13 fractions, TCA-precipitated and the pellets resuspended in Laemmli buffer and analyzed by immunoblotting.

**Copper measurements**. Determination of copper in COX11- or PET191-containing metallochaperone modules was performed by using immunoprecipitates of these proteins and inductively coupled plasma mass spectrometry (ICP-MS). For sample preparation, we used 16 mg of mitochondria purified from the following cell lines: HEK293T WT or COX11-KO cells stably expressing WT or cysteine mutant versions of COX11-FLAG, COX11-KO cells stably expressing PET191-FLAG, and PET191-KO cells stably expressing PET191-FLAG. Mitochondria were solubilized in 0.4% DDM to extract COX11 or PET191 in native conditions, as described earlier. Then, we performed a large-scale immunoprecipitation assay using anti-FLAG-conjugated sepharose beads, and the metals were extracted from the immunoprecipitate by protein denaturation with 1% SDS. The samples were lyophilized, incubated overnight with 100 μl of nitric acid at 65 °C, diluted 20-fold, and metals in the samples were measured using ICP-MS as described previously[67]. Briefly, the samples were digested in 100 μl of 70% w/v trace metal-grade nitric acid for 14 h at 65 °C. The digests were cooled down to room temperature, centrifuged, and diluted 20-fold into the autosampler of the ICP-MS, with a 2% nitric acid spiked with 50 μg/L Gallium as internal standard (final concentration). The instrument consists of an Agilent 7500cx (Santa Clara, CA) operating in Mix reaction mode (3.5 ml $H_2$ and 1.5 ml He per min) to remove polyatomic interferences in the collision cell. The samples were loaded using an autosampler (ESI, Omaha, NE) using a 6-port injection valve with a 125 μl sample loop. The samples were injected at a flow rate of 55 μl/min with the following operating conditions for the ICP-MS: Ar carrier flow, 1.0 L/min; Ar make-up flow, 0.1–0.2 L/min; forward power, 1,500 W; Ar plasma gas, 15 L/min, Ar auxiliary gas, 1 L/min. Concentrations were calculated using an external calibration curve, with 50 ppb Ga as the internal standard. All solutions for metal analysis used metal-grade water and nitric acid. For comparisons among samples from different cell lines, the total Cu concentration data was normalized by [$^{34}$S] and expressed as Cu/S ratio.

**Oxidative-stress measurements**. Reactive oxygen species (ROS) measurements were performed by flow cytometry, using three different dyes, DHE (2.5 mM) and MitoSOX (2 mM) to preferentially detect superoxide anion, and CM-$H_2$DCFDA (5 mM) to preferentially detect hydrogen peroxide. Cells were grown in a 6-well plate and incubated with the dye for 30 min. Then, the cells were washed in PBS, resuspended in Hanks solution, and analyzed using a BD LSR II flow cytometer. We used cells treated for 1 h at 37 °C with 100 μM $H_2O_2$ or 1 μM antimycin A as a positive control

**Cytotoxicity assay**. The cell sensitivity to oxidative stress was assessed colorimetrically with the Cytotoxicity Detection LDH Kit (Sigma, 11644793001). WT and KOs cells were grown to 80% confluency in 6-well plates. Following aspiration of the medium, the cultures were carefully washed with PBS and incubated in FBS-free medium supplemented with 0, 300 or 500 μM $H_2O_2$ for 7 h. One hundred μl of medium was used per assay. LDH activity was measured according to the manufacturer's instructions.

**Yeast strains and growth conditions**. Yeast strains used in this work were of the W303 genetic background (Supplementary Table 3). The cox11Δ, pet191Δ, pet117Δ, coa3Δ, cox11Δcoa3Δ, cox11Δpet191Δ, and cox11Δpet117Δ strains were generated in vivo using homologous recombination of PCR-amplified gene-specific knock-out cassettes containing HIS3, URA3MX, or KanMX4 selection markers flanked by DNA sequences with homology to the upstream and downstream chromosomal regions of the coding sequence for the deleted gene. The plasmid for wild-type Pet191-6xHIS (containing the hexa-His epitope) expression was

generated by PCR amplification of PET191 from genomic DNA followed by restriction enzyme cloning into the pRS426 vector containing the MET25 promoter and CYC1 terminator. Depending on the experiment, yeast cells were cultured in yeast extract-peptone (YP) or synthetic complete (SC) media lacking nutrients (amino acids or nucleotides) necessary to maintain plasmid selective pressure[68] containing either 2% glucose, 2% galactose, or 2% glycerol/2% lactic acid mix as the carbon source. Growth tests to assess respiratory capacity and quantify hydrogen peroxide sensitivity were carried out as before[69,70]. Cultures used for growth tests were grown overnight in SC media lacking relevant nutrients to maintain plasmid selection, then normalized to OD$^{600}$ of 1 and spotted onto SC plates with or without 1 mM CuSO$_4$.

For yeast/human COX11 heterologous complementary studies, yeast strains were grown in the following standard culture media: YPD (2% glucose, 1% yeast extract, 2% peptone), YPEG (2% ethanol, 3% glycerol, 1% yeast extract, 2% peptone) and SC-EG (2% ethanol, 3% glycerol, 0.67% yeast nitrogen base without amino acids) as described[71]. Strains grown in liquid and solid media were incubated at 30 °C. The ScCOX11 gene was amplified from genomic DNA and cloned into pGML3 plasmid using restriction sites BamH1 and HindIII. The HsCOX11 gene was amplified from COX11 cloned in pCMV6 plasmid (OriGene) and sub-cloned into pGML4 plasmid using restriction sites Kpn1 and Sal1. The primers used are listed in the Supplementary Table 2. The yeast strain cox11Δ was reconstituted either with yeast ScCOX11 in pGML3 (LEU resistance and fused to a C-terminal hemagglutinin HA-tag) or human HsCOX11 in pGML4 (URA resistance). Mitochondria were purified from yeast cultures grown in liquid YPGal media (2% galactose, 1% yeast extract, 2% peptone) to mid-exponential phase as described[72].

**Assays in Saccharomyces cerevisiae**. To assess hydrogen peroxide sensitivity of yeast cells of interest, cells were cultured to mid-log phase, normalized, and acutely treated with 1 mM $H_2O_2$ for 1 h at 28 °C. Following treatment, cultures were diluted to 300 cells per sample and plated to assess viable colony forming units after 48 h of growth at 28 °C. Cells were examined in 7 biological replicates, and obtained experimental values were statistically analyzed by one-way ANOVA and Student's t-test.

For immunoblot analysis of yeast mitochondria, mitochondria-enriched fractions were isolated using established protocols[73]. Mitochondrial proteins were separated by SDS-PAGE and transferred to nitrocellulose membrane, blocked in 5% non-fat milk in PBS with 0.1% Tween-20, and incubated with relevant primary antibodies and goat anti-mouse or goat anti-rabbit horseradish peroxidase-coupled secondary antibodies (Santa Cruz Biotechnology) at dilutions 1:10,000. Proteins of interest were visualized by incubation of said membranes with chemiluminescence reagents (Thermo Scientific) and exposure to X-ray film. The following primary antibodies were used: mouse anti-porin (Dilution 1:1000; Thermo Scientific 459500) and rabbit anti-Pet191 (dilution 1:500; kindly provided by Dr. Agnieszka Chacinska, IMol Institute, Warsaw, Poland; Ab reported in ref. [74]). For yeast/human COX11 heterologous complementary studies, mitochondria isolated from cox11Δ yeast strains expressing yeast or human COX11, were resuspended in Laemmli buffer and analyzed by immunoblotting using antibodies again HA (dilution 1:1000; Invitrogen 71-5500), chicken anti-yeast Cox11 (dilution 1:5000; gift from Dennis Winge (University of Utah) reported in ref. [36]), and rabbit anti-human COX11 (OriGene, TA323960). All antibodies were tested for reliability to ensure specificity of detection.

**Statistical analysis**. Unless indicated otherwise, all experiments were performed at least in biological triplicates and results were presented as mean ± standard deviation (SD) or standard error of the mean (SEM) of absolute values or percentages of control. For treatments with copper or elesclomol, for each cell line, the multiple plates used where randomly selected without applying any particular method, for treatment or no treatment. Statistical p values for comparison of two groups were obtained by applying a Student's two-tailed unpaired t-test. For comparison of multiple groups, performed one-way analysis of variance (ANOVA) followed by Tukey's post-hoc test for all the groups, using the GraphPad Prism 9.3.1. A $p < 0.05$ was considered significant test. Exact p values are indicated in the graphs, except for $p < 0.0001$ that are denoted as ****. Data in X-ray films were digitalized and analyzed with the ImageJ software version 1.53r or the histogram panel of Adobe Photoshop.

**Reporting summary**. Further information on research design is available in the Nature Research Reporting Summary linked to this article.

# Data availability
All unique/stable reagents generated in this study (plasmids and cell lines) are available from the corresponding author with a completed Materials Transfer Agreement. Proteomics data have been deposited in PRIDE[75] as PXD034576, PXD034578, PXD034579, and PXD034577. The ICP-MS data has been uploaded to Mendeley data: Nyvltova, Eva (2022), "Nyvltova et al. ICP-MS Data", Mendeley Data, V1, doi: 10.17632/w2yk8g568t.1 and can be accessed at https://data.mendeley.com/datasets/w2yk8g568t/1. Source data are provided with this paper.

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

## Acknowledgements

We thank Drs. Flavia Fontanesi and Carlos T. Moraes for critical reading of the manuscript. We thank Drs. G. Manfredi, J.A. Enriquez, M. Ryan, and P. Rehling for providing reagents. This research was supported by the National Institutes of Health grants R35-GM118141 (A.B.), R35-GM131701-01 (O.K.), and T32-GM107001 (J.V.D.). Muscular Dystrophy Association grant MDA-381828 (A.B.). American Heart Association postdoctoral fellowship 19POST34430019 (E.N.).

## Author contributions

Conceptualization: E.N., A.B. Methodology: E.N., O.K., A.B. Investigation: E.N., J.V.D., J.S., O.K., A.B. Funding acquisition: E.N., O.K., A.B. Writing—original draft: E.N., A.B. Writing—review and editing: E.N., J.V.D., J.S., O.K., A.B.

## Competing interests

The authors declare no competing interests.
