## [Peer Review File · Nature Communications]

Coordination of metal center biogenesis in human cytochrome c oxidaseREVIEWER COMMENTS

Reviewer #1 (Remarks to the Author):

Key Results:

Mammalian cytochrome c oxidase (CcO) has two copper (Cu) centers; (1) CuA in COX2 and (2) CuB in COX1. Dysregulation of Cu-binding to these centers can result in the production and accumulation of reactive intermediates that can disrupt physiological cellular function. While the assembly of the CuA site in COX2 is well characterized, CuB assembly in COX1 is less understood. Here, the authors aim to characterize the process of CuB assembly through various biochemical assays.

Validity:

The authors have done a significant amount of work using immunoprecipitation and immunoblotting to show how expression levels of certain components of the cytochrome c oxidase assembly effect one another as well as surrounding proteins involved in copper coordination.

Some of the conclusions drawn from these assays are lacking more direct, supportive data that substantiate these claims.

For example, in the first results section 'COX11 is not essential for CcO biogenesis in human mitochondria' this conclusion is drawn from densitometry analysis from immunoblotting and ImageJ software to show decreases in protein expression from COX11-KO cell lines and mitochondrial cytochrome spectra both of which assays can suffer from high variability. I have included some specific comments about the mitochondrial spectral data.

I am concerned that the conclusion that 'COX11, COX19 and PET191 are functionally linked to COX2 metalation' drawn from the results section 'COX19, PET191, COA6 and COX2 are essential to maintain the redox state and copper binding capacity of COX11' is overstated from the results generated. Could the authors take a more approach in these KO cell lines to demonstrate that copper is or isn't bound to COX11 and show the presence or absence of AMS binding as a result? Perhaps by inorganic LC-MS/MS to show metal status as well as peptide identification of COX11 and AMS presence/absence?

Data, methodology & suggested improvements:

Page 5, Line 118 – Authors indicate that α peak corresponding to hemes a+a3 of approx. 75% of WT. How is this calculation performed? Can this calculation be made with given analysis lacks quantitative standards or data required to make this type of estimate? This statement doesn't reference Figure 11.

Page 5, Lines 120-122 – Authors speculate that a spectral shift of 5-6 nm in the α peak indicates changes in the environments surrounding the heme centers in COX1. As this data appears to be an n of 1 for each background how can they be sure it's not an effect of instrument drift or part of the standard deviation intrinsic to the analytical technique? If raw data was provided of multiple measurements showing the robustness of the result, then the author's conclusions could be sounder.

Page 5, Lines 123-124 – Similar to above comment, from this spectral shift the authors imply this result demonstrates COX11 and COX19 are not involved in heme α biosynthesis and delivery to COX1. However, the data does not seem sound enough to make this claim.

Page 5, Lines 128-130 – Authors indicate that the media their cells were grown in was supplemented with copper, 'Culture media supplementation with Cu(I), His-Cu(II), or elesclomol-Cu(II)...' However, was the copper content in the media analysed to ensure copper levels were elevated to support their claim that elevated levels of copper did not enhance CcO activity in COX11-KO cells? Also, could not find mention of this in methods section. How much of each copper source was used to supplement media?

Page 7, Line 197-199 – Authors indicate that SCO2 and COX22 are interacting with apo-COX2 however the metal status of COX2 has not been verified.

Page 8, Line 220 – Analysis of interacting proteins from IP is said to be done by LC-MS/MS for which the author's indicate confirmation of protein-protein interactions displayed in Figure 3C. However, the authors do not include any of the methods or raw data from the LC-MS/MS analyses to support the results in Figure 3C.

Page 9, Line 241 – Authors indicate their data 'containing unmetallated Cu-chaperones' however the metal status of these chaperones has not been reported.

Page 10, Line 290 – Authors state that from ICP-MS analysis complexes do not contain 'appreciable quantities of bound copper.' Can this be further elaborated as some results indicate approx. 60% of WT but without molar ratios of protein to metal it is hard to know if this result is due to non-specific binding or an actual loss of coordination by the cognate metal ions.

Page 14, Line 404 – (F-H) should be in bold font as well as (I) on the following line.

Page 19, Line 543 – There might be some words missing to complete the sentence ‘...same volume of the vehicle 100% ethanol.’

Page 21, Line 605 – first mention of DMEM, worth writing out entire chemical name for consistency.

Page 21, Lines 610, 611 & 614 – EDTA, PMSF & SDS have already been introduced therefore writing their entire chemical names here is un-necessary.

Page 23, Line 675 – ‘ we ’ is un-necessary

Page 24, Line 691 – Remove ‘) ’ after Ar carrier flow

When reporting the COX11-AMS conjugation to demonstrate copper binding did the authors consider using LC-MS/MS to confirm AMS since this technique was mentioned as part of other protein-protein interaction confirmations?

Did the authors normalize the amount of protein being analyzed by ICP-MS? Given the way the copper data is reported it may be unclear whether some of these changes are due to differences in final protein concentration immediately prior to sample preparation for analysis. I can see in the methods section that a starting amount of 8mg of mitochondrial extract was used but the final amount of protein following sample preparation was not reported.

The authors report several times in the results section that medias were supplemented with copper (Page 5, Lines 128 onward; Page 6, Line 141 & 146) but do not report the amount of copper supplemented, direct copper measurements of media or cells cultured in media to show elevated copper and the details regarding copper supplementation in Fig 1G are not clear.

Clarity and context:

The Abstract and Main sections are written very well and easy to read and set up the experimental aims well.

Throughout the Methods section there are inconsistencies with units for example:

- (1) MgCl₂ vs MgCl₂ (Page 19, Line 537 & Page 18, Line 532, respectively)
- (2) 4 °C vs 4°C (Page 22, Line 635 & Page 23, Line 668, respectively)
- (3) pH 7,4 vs pH 7.4 (Page 22, Line 632 & Page 19, Line 537, respectively)
- (4) 1mM vs 20 mM (Page 23, Line 665 & Line 667, respectively)

Reviewer #2 (Remarks to the Author):

In this comprehensive study, Nyvltova et al., have attempted to elucidate the complex process of biogenesis of cytochrome c oxidase (CcO), a multimeric membrane protein of mitochondrial electron transport chain. The authors have used a combination of KO cell lines of assembly factors of CcO and in vivo biochemical assays to determine the sequence of multi-step process of building CcO holoenzyme containing its cofactors – copper and heme. The authors mainly focused on proteins involved in copper delivery to CcO such as COX11 and COX19 and show that although the proteins that form the copper sites are distinct, the assembly of these copper sites is interlinked. The authors propose this as a possible regulatory mechanism that prevents reactive oxygen species generation from partially assembled intermediates that might accumulate if the assembly of one of the copper site is disrupted. The authors also raise questions about the existing knowledge in the field – that COX11 is the copper donor for Cu_A site. The scope of the study is very broad resulting in a very descriptive work, some of which is contradictory to existing knowledge. Moreover, the manuscript needs to be thoroughly proofread, as there are number errors in data listing and interpretation. There are many additional concerns, some of which are listed below:

Major concerns

1. The data rich paper reports a lot of interactions, which forms the basis of the model shown in Fig. 6. However, the data can be interpreted in multiple ways and different models could be predicted. Some

streamlining will help deliver the main message. For example, data from figure 4 does not add to the main message of the paper and could be removed.

2. Some of the statements do not match the data. For example, in line 137-138 the authors state “overexpression of COX19 in the COX11-KO cell line did not improve its residual activity and vice versa,” however there is no data for the “vice versa” part of the statement. Similarly, the statement in line 141, even in the presence of exogenous Cu (Fig. 1F) and line 146 (Fig. 1H) there is no indication of exogenous copper in the figure.

3. The authors should show the levels of COX1 in Fig 1E.

4. The decrease in most of the CcO biogenesis factors observed in COX19KO and PET191KO shown in Supplementary Fig. 2A and B could be simply due to decreased mitochondrial mass. Instead of using Actin as a loading control the authors should use other mitochondrial proteins such as VDAC, an inner mitochondrial membrane protein and a mitochondrial matrix protein. This will tell whether the effect of the loss of COX19 and PET191 is specific to CcO biogenesis factors or is common to other mitochondrial proteins.

5. Since COX11 has been previously shown to participate in CuB site formation, the data showing its more prominent role in CuA site formation are really surprising. The author should provide an explanation or more experimental data to support this observation. Is it possible that COX11 function has diverged from unicellular eukaryotes to metazoans? The authors should test whether human COX11 can replace yeast COX11KO or vice versa. The results from this experiment could be more meaningful than sequence based prediction of COX11 function.

6. Another example of data not matching the text is in Supplementary Fig. 3, where the authors state “PET191....., weakly suppresses the respiratory growth defect of a COX11 strain” The statement is incorrect as there is no rescue in YPGL respiratory media. There is a slight rescue in SCGal or YPGal fermentative media, though severe growth defect of COX11 KO in this media is surprising. Is growth defect of COX11KO in YPGAL media previously reported? The authors should confirm if this phenotype is indeed a result of COX11KO and not due to some other mutation in the strain background by transforming back with COX11 plasmid and showing full rescue of growth in YPGal.

7. Figure 1H and Supplementary Figure 2F are identical and one of them should be removed.

8. Some of the data are internally inconsistent. For example, there is discrepancy between Fig. 1E and 1F, where SCO1 overexpression in COX11KO results in increased levels of COX1 (Fig. 1E) but not the CcO activity (Fig. 1F).

9. In line 157-158, “mutation abolished PET191 capacity to suppress the COX11-KO CcO deficiency” is misleading because even the WT PET191 overexpression failed to rescue COX11 KO (Supplementary Fig. 4F compare lanes 2 and 4). Also, it is not clear why supplementary figure 4G is cited to describe this result.

10. The graphical description of reverse thiol-trapping approach should be improved. For example, the authors could provide the structures of the products at each step in Fig2A instead of just providing the structure of IAA or AMS. This can be illustrated in the form of three examples depicting the following

scenarios: All three cysteines are reduced and free, one free cysteine and a disulfide, one free cysteine and two copper bound cysteines. Fig 2B is misleading and should be removed.

11. The interpretation of reverse thiol trapping data is misleading. For example, in line 170, "In WT HEK293T cells 50% of COX11 is Cu(I).COX11" Treatment with just AMS (last lane in fig 2C) should give us details about the cysteines that are not oxidized i.e., free and copper bound cysteines should react with AMS. However, in Fig 2C only one cysteine reacted with AMS indicating that the other two are oxidized and not copper bound. In that case, the authors should clarify why they say that in WT, 50% of COX11 is copper bound. Similar problems are seen in fig 2E.

12. In lines 172-173 (Fig. 2D) "only apo-COX11 was detected in COA6-KO" the data from lanes 3 and 4 do not add up.

13. In line 183-184, the statement that "COX11 exhibits a stable interaction with newly synthesized COX2 but not with COX1 in WT HEK293T cells" is not accurate. There is an enrichment of signal for COX1 in HEK293T cells (though it is less than COX2) (Fig. 3A).

14. Supplementary Figure 5A do not show that interaction persists between COX11 and COX2 in the absence of COX1 as stated in line 185.

15. The heatmap in Supplementary Fig. 5B do not match the statement in line 192-193.

16. In lines 193 -194, the authors state that SCO1, COA6 and COX16 failed to interact with COX2 in the absence of PET191. In Supplementary Fig 2B, the authors have shown that in PET191-KO the levels of SCO1, COA6 and COX16 are already severely down. So how did the authors account for these decreased levels in the interpretation of their IP data? The authors should also provide the levels of immunoprecipitated protein in Supplementary Fig 5B as controls.

17. In lines 196-197 the authors state that the interaction of COX11 with COX2 is independent of PET191 or COX19. However, in Supplementary Fig 5B, the COX11 IP did not show any co-IP of COX2 in WT itself in both cases. In that case, how did the authors come to this conclusion? Also, this contradicts their data in Fig 3A that COX11 interacts with COX2. This further raises questions about the reproducibility of the results from this study.

18. The authors should explain how they normalized the data in figure 5A.

19. In lines 297-300 the authors should explain why they say PET191 binds copper in the absence of COX11. ICP-MS of the cross-linked interactome is not a clear way to see if a protein binds copper or not.

Minor concerns

1. In lines 285 and 288 Fig.3D should have been Supplementary Fig 6H.

2. Each blot in a figure should be considered a separate panel and should be referenced in the text appropriate for ease of understanding - ex: Supplementary Fig 5B can be split into 8 panels.

3. What is WT-NT in Fig 5A? Explain in the figure legend

4. Explain the abbreviations Ex, Un used in Fig 3 in the figure legend.

5. Why is COX1 not included in Fig 3C?

Reviewer #3 (Remarks to the Author):

The study by Nývltová investigates the roles of COX11, COX19 and PET191/COA5 in COX assembly in metazoans. The functional characterization of these three factors outside of budding yeast has been lacking, and is therefore a very worthwhile avenue of investigation and one whose advancement is crucial to a fuller understanding of how the holoenzyme is made. An impressive amount of data was collected and is presented but the manuscript in its current construction is very difficult to digest. Moreover, there appears to be somewhat of a disconnect between the emphasis placed on the content of the Title and the Abstract relative to the central findings described by the authors. More specifically, it has been known for some time that stalled assembly intermediates relating to maturation of early assembly modules leads to the production of cytotoxic reactive intermediates (e.g. Khalimonchuk et al. 2007, Veniamin et al. 2011). In contrast the Abstract states the central finding as “Here, we report that in human cells the CcO copper chaperones form macromolecular assemblies and cooperate with several twin-CX9C proteins to control heme a biosynthesis and coordinate copper transfer sequentially to the CuA and CuB sites.”. Although too strongly worded in this Reviewer’s opinion (see below for more detail), this is where the real novelty of the work under consideration is found. My specific comments, questions and/or criticisms can be found below.

1. Fig 1A/Fig S2A,B: There is no doubt that the KO lines are authentic, and I appreciate how thoroughly they were characterized before downstream experiments were conducted. However, actin is a loading control to ensure equal amounts of whole cell extracts were loaded across lanes and I would like to see a mitochondrial loading control (e.g. TOM20/40) to ensure that organelle content is comparable when comparing WT cells to knockout and functionally complemented lines.

2. Fig 1C/Fig S2C,D: What is “sub CIV”? Assembly intermediates have traditionally been viewed as S1-S3, with S2 being a prominent intermediate seen when there is a failure to mature the CuA site of COX2. If S2 is in fact accumulating in COX11 KO (and COX19 KO and PET191 KO) cells, it would be consistent with your other findings that COX11 interacts with COX2 and is crucial for its maturation.

3. Fig 1H/Fig S2F: The finding that PET191 overexpression can partially complement the COX11 KO cell lines and that its Cys residues are required (Fig 1F) is an important one. Why, however, have the authors duplicated identical data in two distinct figure panels? This is worrisome.

4. Fig 1I: The relative abundance of heme a/a₃ based on the Soret band argues that COX11 and COX19 are not required for hemylation of COX1, which is a really neat finding. Why, however, does the peak not return to 603nm upon overexpressing PET191 in the COX11 KO? This suggests that even though you have an increase in enzyme activity and COX2 abundance (Fig 2E, S4E), that the binuclear center may not be in its native state. Consistent with this idea, no complementation of native enzyme levels are observed by BN PAGE (Fig S4F). Reconciling these findings is important yet isn't raised in either the Results or Discussion.

5. Fig 2: The experiments quantifying the redox state of the cysteinyl sulphurs of COX11 are very convincing. My one question for the authors is how do they discriminate between a direct versus an indirect effect of deleting COX19, PET191 or COA6 on the Cys redox state of COX11? More specifically, given that we know this will lead to the accumulation of stalled assembly intermediates how can the authors rule out the possibility that the disproportionate oxidation of cysteinyl sulphurs in COX11 isn't attributable to enhanced radical production and therefore changes in the local redox environment? It would seem an experiment where COX10 is silenced as was done in Fig 5C would be an important one to discriminate between a direct and indirect effect of loss of these three COX assembly factors on the redox regulation of COX11.

6. Fig 3C: The finding that COX11 interacts with newly synthesized COX2 is convincing and very novel. This Reviewer is having trouble, however, rationalizing why fewer interacting partners for COX11 and PET191 were identified using MS than IP/WB analysis, given that the former approach is infinitely more sensitive. A gold standard when using physical methods to establish that two proteins interact is a reciprocal IP. Even though this would be an onerous task given the sheer number of IPs conducted in this study, it seems crucial since the manuscript as a whole aims to build a model (Fig 6) that describes how subassembly intermediates are temporally populated with ancillary factors as a given module is matured and merged with others to build the native enzyme.

7. Figs 2, 4 and S4: Panels A and B argue that PET191 retains some residual function in the absence of its conserved cysteines. However, based on Fig S4F it acts as a dominant negative when expressed in a COX11 KO background, further reducing residual COX content. Given the pronounced effects of loss of PET191 on the redox state of the cysteinyl sulphurs of COX11 (Fig 2C), SCO1 (Fig 4B) and SCO2 (Fig 4G), all of which become more oxidized, it would seem important to determine whether overexpressing mutant PET191 exerts similar or different redox effects on these three factors. Again, with the current data it is impossible to discriminate between the possibility that PET191 acts directly on COX11, SCO1 and/or SCO2 to modulate their redox state or indirectly by serving as a scaffold protein/stabilizing factor that regulates the activity of another thiol oxidoreductase.

8. Fig 5A: The figure legend says the copper content of the metallochaperone modules was quantified yet the methods describe that the analysis was conducted on whole mitochondria. If the latter is in fact what was used as starting material, these results cannot discriminate where the copper itself is found.

For e.g., it could be part of the labile pool housed in the matrix, it could be bound to metallochaperones or to target proteins (COX1, COX2). This Reviewer only sees a significant increase in copper upon overexpressing COX11 or PET191 with all other conditions being comparable to WT NT. No data is shown for COX11 KO or PET191 KO cells. Thus, there is a limited ability to draw firm conclusions from this analysis, especially since the authors are trying to temporally link hemylation and copper insertion to specific stages of holoenzyme assembly. A more arduous but rigorous approach would be to IP the MCC and earlier subassembly intermediates (Fig S5D argues they can do this), fractionate them by size exclusion under native conditions using an HPLC that is coupled to an in-line MS spectrometer (e.g. Soma et al. 2019). This would allow you to definitively link a given metal to a specific stage of holoenzyme biogenesis.

9. The Discussion: The real novelty of this manuscript is its advancement of our understanding of COX11, COX19 and PET191/COA5 function. Yet many holes remain in that knowledge. Thus, this Reviewer found the Discussion to be focused too much on building a holistic model of holoenzyme assembly that overstretched the interpretation of the data in hand with respect to when and by whom the copper sites are formed to allow modules to merge and holoenzyme assembly to proceed. While not much is known about COX11, COX19 and PET191/COA5 function in metazoans, some uncited studies have looked at these proteins and how they may interact with other COX assembly factors to promote holoenzyme assembly. A significant rewrite is warranted.

MINOR:

1. Fig S3D needs better labelling and/or a clearer description in the legend given that at present the distinction between the top pair and bottom pair of graphs is hard to understand.

2. Pg 10 “The entrapment of COX10 in complexes with CuA and CuB chaperones until their Cu loading reflects a concerted regulation of metalation of all the redox centers in CcO.” I cannot find any direct evidence that is presented to tie it metallation, particularly of the copper centers.

3. Pg 13 “We find that contrary to the established dogma, COX11 is important but dispensable in metazoans.” This statement is very strong given the finding was generated in a model human cell line cultured under supraphysiological conditions and suggests that humans lacking functional COX11 would be free of disease.

Response to Reviewers

Reviewer #1:

Key Results:

Mammalian cytochrome c oxidase (CcO) has two copper (Cu) centers; (1) CuA in COX2 and (2) CuB in COX1. Dysregulation of Cu-binding to these centers can result in the production and accumulation of reactive intermediates that can disrupt physiological cellular function. While the assembly of the CuA site in COX2 is well characterized, CuB assembly in COX1 is less understood. Here, the authors aim to characterize the process of CuB assembly through various biochemical assays.

Validity:

The authors have done a significant amount of work using immunoprecipitation and immunoblotting to show how expression levels of certain components of the cytochrome c oxidase assembly effect one another as well as surrounding proteins involved in copper coordination. Some of the conclusions drawn from these assays are lacking more direct, supportive data that substantiate these claims.

For example, in the first results section 'COX11 is not essential for CcO biogenesis in human mitochondria' this conclusion is drawn from densitometry analysis from immunoblotting and ImageJ software to show decreases in protein expression from COX11-KO cell lines and mitochondrial cytochrome spectra both of which assays can suffer from high variability. I have included some specific comments about the mitochondrial spectral data.

Thank you for pointing this out. The conclusion that "COX11 is not essential for CcO biogenesis in human mitochondria" was actually drawn from several data sets. Most importantly, it is premised on the finding that compared to WT, COX11-KO cells assemble 20% functional CcO enzyme fully able to oxidize cytochrome c (**Fig. 1D**), and support 60% of respiratory capacity (**Fig. 1E**). In agreement with the functional data, residual steady-state levels of CcO core subunits COX1 and COX2 (**Fig. 1A**), and fully assembled CIV and Complex IV-containing supercomplexes (**Fig. 1C**) separated by SDS-PAGE and BN-PAGE respectively, were detected by immunoblotting. Therefore, we believe that our data firmly support the conclusion that CIV is still present and functional in COX11-KO cells. Nevertheless, for better clarity, the title of the respective section has been changed to "COX19 is essential for human CcO biogenesis, but the absence of COX11 allows some residual CcO assembly and function".

I am concerned that the conclusion that 'COX11, COX19 and PET191 are functionally linked to COX2 metalation' drawn from the results section 'COX19, PET191, COA6 and COX2 are essential to maintain the redox state and copper binding capacity of COX11' is overstated from the results generated. Could the authors take a more approach in these KO cell lines to demonstrate that copper is or isn't bound to COX11 and show the presence or absence of AMS binding as a result? Perhaps by inorganic LC-MS/MS to show metal status as well as peptide identification of COX11 and AMS presence/absence?

The reviewer is raising two important points in their comment. First, we have concluded that 'COX11, COX19 and PET191 are functionally linked to COX2 metalation' because the *in cellulo* redox state of COX11 is modified by the absence of COX2 or its metallochaperone COA6. At

that point, we are not disclosing whether the link is direct or indirect. In the following section, we present data demonstrating that COX11 interacts with newly synthesized COX2. In the subsequent sections, we show that COX2 is unstable in the absence of COX19 or PET191, as it is synthesized but has undetectable steady-state levels. These two proteins create a stable complex with COX2 metalation chaperones SCO1 and SCO2 and the copper donor chaperone COX17.

The second point pertains to the analysis of copper binding to COX11. This is a technically challenging experiment to perform *in cellulo* but we have used inductively coupled plasma MS (ICP-MS) elemental analysis and obtained data showing that immunoprecipitated COX11 indeed binds copper (**Fig. 5D**). The AMS binding-based reverse thiol-trapping assay does not distinguish copper coordination by COX11 using two cysteines vs. an intramolecular disulfide bond. However, it has been shown in bacteria that native monomeric COX11 does not form an intramolecular disulfide, but only dimerizes to coordinate two copper molecules intermolecularly and not to form intermolecular disulfides ¹. The existence of a monomeric oxidized COX11 species has been posited only in yeast based on direct thiol trapping assays ², wherein copper binding to COX11 was not considered for data interpretation. Incidentally, we now present data that in human mitochondria, the oligomerization state of COX11 does not strictly depend on its redox or copper binding state (new **Fig. S6**). As indicated above, the thiol trapping assays cannot directly distinguish between intermolecular copper coordination and intramolecular disulfide bond. However, all the available data supports the former notion, including our data indicating that only reduced COX11 is detected in the absence of COX19, PET191, COA6 or COX2. Therefore, we believe that our thiol trapping assays are informative. The use of LC-MS/MS suggested by the reviewer would not be informative in this case since the tryptic peptide that is generated including a relevant cysteine (CSLR) is too short to be detected by most standard LC-MS/MS techniques.

Data, methodology & suggested improvements:

Page 5, Line 118 – Authors indicate that α peak corresponding to hemes $a+a_3$ of approx. 75% of WT. How is this calculation performed? Can this calculation be made with given analysis lacks quantitative standards or data required to make this type of estimate? This statement doesn't reference Figure 1I.

Quantification of cytochromes has been included in the updated Figure 1 (the new **Fig. 1F**), which is now referenced, and the method used explained in the methods section. Briefly, the heights of the peaks and the area under the peaks in at least three independent experiments were calculated using the of the Quant mode of the UV-Probe software (Shimadzu) and expressed as the $a+a_3/b$ ratio. Similar values were obtained when using the peaks height and the area under the peaks.

Page 5, Lines 120-122 – Authors speculate that a spectral shift of 5-6 nm in the α peak indicates changes in the environments surrounding the heme centers in COX1. As this data appears to be an n of 1 for each background how can they be sure it's not an effect of instrument drift or part of the standard deviation intrinsic to the analytical technique? If raw data was provided of multiple measurements showing the robustness of the result, then the author's conclusions could be sounder.

Experiments were carried out in triplicate, using three independent mitochondrial preparations for every cell line analyzed. Each sample was then scanned three times to account for any potential "instrument drift". Only one representative set of spectra is shown in **Fig. 1F**, but additional spectra from replicate measurements are now included in the **raw data** file.

Page 5, Lines 123-124 – Similar to above comment, from this spectral shift the authors imply this result demonstrates COX11 and COX19 are not involved in heme α biosynthesis and delivery to COX1. However, the data does not seem sound enough to make this claim.

The new quantification data added to **Fig. 1F** and **raw data** from several repetitions of this experiment, show that some heme $a+a_3$ peaking at 597nm can be reproducibly detected in the COX11-KO and COX19-KO mitochondrial extracts. Specifically, the heme $a+a_3$ /heme b ratio was ~75% of WT levels for COX11-KO extracts, and ~20% of WT levels for COX19-KO extracts. To avoid any confusion, we have changed the sentence to: "...These data imply that heme a biosynthesis and delivery to COX1 can proceed independently of COX11 and COX19".

Page 5, Lines 128-130 – Authors indicate that the media their cells were grown in was supplemented with copper, 'Culture media supplementation with Cu(I), His-Cu(II), or elesclomol-Cu(II)...' However, was the copper content in the media analysed to ensure copper levels were elevated to support their claim that elevated levels of copper did not enhance CcO activity in COX11-KO cells? Also, could not find mention of this in methods section. How much of each copper source was used to supplement media?

For all cell culture assays we used regular DMEM-based medium that contains all essential components including copper. For copper supplementation experiments, the medium was supplemented with additional (i) 1.5 mM Cu-His, (ii) 1 mM CuCl₂, or (iii) or 1 nM elesclomol + 1mM CuCl₂. This information has been included in the respective figure legends and method sections.

Page 7, Line 197-199 – Authors indicate that SCO2 and COX22 are interacting with apo-COX2 however the metal status of COX2 has not been verified.

The actual statement was that "... SCO2 and COX11 interact with apo-COX2 before the action of PET191 and COX19, and prior to the joining of SCO1, COA6, and COX16." It is true that we did not verify the metal status of COX2 at that stage. However, it is well established in the literature that COX2 is metalated by the copper chaperone SCO1. Prior to the COX2-SCO1 interaction, COX2 is commonly believed to remain unmetalated.

Page 8, Line 220 – Analysis of interacting proteins from IP is said to be done by LC-MS/MS for which the author's indicate confirmation of protein-protein interactions displayed in Figure 3C. However, the authors do not include any of the methods or raw data from the LC-MS/MS analyses to support the results in Figure 3C.

Thank you for pointing this out. The methods for the LC-MS/MS analyses have now been included. The raw data are published at Mendeley data in Barrientos, Antonio (2021), "Nyvltova et al. MS Data", Mendeley Data, V1, doi: 10.17632/7jsmsm3xrn.1

Page 9, Line 241 – Authors indicate their data 'containing unmetalated Cu-chaperones' however the metal status of these chaperones has not been reported.

The specified text discusses the formation of a complex containing several copper chaperones and PET191 but not COX17. We are hypothesizing that PET191 may act as a placeholder for COX17. Because COX17 is a well-established Cu donor for SCO1 and COX11, its absence in that complex strongly suggests that metallochaperones in the complex remain in an apo-state.

In support of this interpretation, our Cu measurements show complexes that form with WT COX11 contain copper, whereas complexes that harbor a COX11 variant with mutated copper-coordinating cysteine residues do not (**Fig. 5D**), despite of containing both SCO1 and SCO2 (**Fig. 3D** and **Supplementary Fig. 10H**).

Nevertheless, to avoid being categoric in our conclusions, we have changed the respective verbiage to “containing presumably unmetalated...”.

Page 10, Line 290 – Authors state that from ICP-MS analysis complexes do not contain ‘appreciable quantities of bound copper.’ Can this be further elaborated as some results indicate approx. 60% of WT but without molar ratios of protein to metal it is hard to know if this result is due to non-specific binding or an actual loss of coordination by the cognate metal ions.

In that sentence, we intended to express that a cysteine variant of COX11 was not able to bind copper. However, since sentence was not clear enough, we have now clarified the message. The paragraph reads now as: “To assess whether the metallochaperone complexes formed in the presence of WT or mutant COX11 contain copper, we extracted them from purified mitochondria in native conditions without using crosslinkers, FLAG-immunoprecipitated the complexes, and analyzed them by inductively coupled plasma MS (ICP-MS) elemental analysis (**Fig. 5C**). ICP-MS revealed the presence of significant amounts of Cu in these stable complexes when containing WT COX11, and the absence of detectable bound Cu in complexes associated with mutant COX11 (**Fig. 5D**).”

In our assays, we normalized the Cu data by the levels of ³⁴S detected in the same ICP-MS assays. We need to consider that we are pulling down complexes and no single proteins. Therefore, to calculate metal-protein molar ratios is not obvious. However, we have added controls to estimate background, non-specific binding, which facilitate data interpretation.

Page 14, Line 404 – (F-H) should be in bold font as well as (I) on the following line.

This has been fixed.

Page 19, Line 543 – There might be some words missing to complete the sentence ‘...same volume of the vehicle 100% ethanol.’

The sentence has been fixed. The missing parentheses were included: ‘...same volume of the vehicle (100% ethanol).’

Page 21, Line 605 – first mention of DMEM, worth writing out entire chemical name for consistency.

The sentence has been modified and DMEM is no longer mentioned.

Page 21, Lines 610, 611 & 614 – EDTA, PMSF & SDS have already been introduced therefore writing their entire chemical names here is un-necessary.

This has been fixed.

Page 23, Line 675 – ‘ we ’ is un-necessary

Page 24, Line 691 – Remove ‘) ’ after Ar carrier flow

The two typos pointed out by the reviewer were fixed.

When reporting the COX11-AMS conjugation to demonstrate copper binding did the authors consider using LC-MS/MS to confirm AMS since this technique was mentioned as part of other protein-protein interaction confirmations?

As indicated in our response to the reviewer's point #2, we have not used LC-MS/MS to confirm AMS binding given the robustness of data obtained by regular immunoblot analyses.

Did the authors normalize the amount of protein being analyzed by ICP-MS? Given the way the copper data is reported it may be unclear whether some of these changes are due to differences in final protein concentration immediately prior to sample preparation for analysis. I can see in the methods section that a starting amount of 8mg of mitochondrial extract was used but the final amount of protein following sample preparation was not reported.

This is a good suggestion. **Fig. 5D** shows copper content in COX11- and PET191-containing metallochaperone modules (or complexes) measured by ICP-MS. For these experiments, mitochondria were purified from *COX11-KO* or *PET191-KO* cells expressing the FLAG-tagged COX11 or PET191 proteins (WT or respective mutant variants). The proteins were immunoprecipitated using anti-FLAG conjugated beads and their metal content assessed by ICP-MS. In the original version of this manuscript, the data were expressed as [Cu] in parts per billion normalized by the background value (anti-FLAG beads incubated with mitochondrial lysates of WT cells not expressing any FLAG-tagged protein). As per reviewer #2 request, we have now presented these data as total Cu normalized by ³⁴S. The relevant specifics of the method have been included in the **Fig. 5** legend as well as the methods section.

The authors report several times in the results section that medias were supplemented with copper (Page 5, Lines 128 onward; Page 6, Line 141 & 146) but do not report the amount of copper supplemented, direct copper measurements of media or cells cultured in media to show elevated copper and the details regarding copper supplementation in Fig 1G are not clear.

We apologize for not being clear on this point. As mentioned in the response to the reviewer's point # 6, for all cell culture assays we used regular DMEM-based medium, which contains all essential components including copper. For the copper supplementation experiments, the medium was further supplemented with additional (i) 1.5 mM Cu-His, (ii) 1 mM CuCl₂, or (iii) or 1 nM elesclomol + 1 mM CuCl₂. This information is now included in the indicated figure legends and method sections.

Clarity and context:

The Abstract and Main sections are written very well and easy to read and set up the experimental aims well.

Throughout the Methods section there are inconsistencies with units for example:

- (1) MgClI vs MgCl2 (Page 19, Line 537 & Page 18, Line 532, respectively)
- (2) 4 °C vs 4°C (Page 22, Line 635 & Page 23, Line 668, respectively)
- (3) pH 7,4 vs pH 7.4 (Page 22, Line 632 & Page 19, Line 537, respectively)
- (4) 1mM vs 20 mM (Page 23, Line 665 & Line 667, respectively)

We have now fixed the stated inconsistencies noticed by the reviewer for noticing.

Reviewer #2:

In this comprehensive study, Nyvltova et al., have attempted to elucidate the complex process of biogenesis of cytochrome c oxidase (CcO), a multimeric membrane protein of mitochondrial electron transport chain. The authors have used a combination of KO cell lines of assembly factors of CcO and in vivo biochemical assays to determine the sequence of multi-step process of building CcO holoenzyme containing its cofactors – copper and heme. The authors mainly focused on proteins involved in copper delivery to CcO such as COX11 and COX19 and show that although the proteins that form the copper sites are distinct, the assembly of these copper sites is interlinked. The authors propose this as a possible regulatory mechanism that prevents reactive oxygen species generation from partially assembled intermediates that might accumulate if the assembly of one of the copper site is disrupted.

The authors also raise questions about the existing knowledge in the field – that COX11 is the copper donor for Cu_A site.

We thank the reviewer for raising this point because it suggests that our original text was not clear enough. It is important to note that the outcomes of our study only challenge one previous assumption. This assumption that COX11 is essential or absolutely necessary for Cu_B site metalation is based solely on studies conducted in bacteria and yeast. To clarify our point, we are not proposing that COX11 is the copper donor for Cu_A site in COX2. Rather, we show that in human mitochondria COX11 forms complexes with COX2 metallochaperones (e.g., SCO1/SCO2), thereby influencing the formation of the Cu_A site. Thus, one of the major and perhaps most exciting points of our contribution is that metalation of Cu_A and Cu_B are coordinated.

The scope of the study is very broad resulting in a very descriptive work, some of which is contradictory to existing knowledge.

We agree with the reviewer in that the scope of the study may appear broad given the large number of proteins involved in the process under study. The goal is, however, well defined on the characterization of the CcO assembly pathway steps leading to the assembly of the copper sites in the core subunits COX1 and COX2. The process is thoroughly described as needed but provides abundant mechanistic insight. As mentioned above, our data does not contradict the previous knowledge. We respectfully believe that our data clarifies concepts and close gaps that existed in the previous knowledge regarding the CcO assembly process in human mitochondria.

Moreover, the manuscript needs to be thoroughly proofread, as there are number errors in data listing and interpretation. There are many additional concerns, some of which are listed below:

We sincerely thank the reviewer for their effort in identifying all potential errors. We have worked towards fixing them and believe that, as a result, this revised version of the manuscript is significantly stronger.

Major concerns

1. The data rich paper reports a lot of interactions, which forms the basis of the model shown in Fig. 6. However, the data can be interpreted in multiple ways and different models could be predicted. Some streamlining will help deliver the main message. For example, data from figure 4 does not add to the main message of the paper and could be removed.

Thank you for pointing this out. We concur with this notion and have done our best to incorporate alternative explanations, but we are also presenting the model that we believe better fits all the available data. After introducing all the revisions requested by the reviewers, we still

believe that Fig 4 is informative enough to be maintained among the main figures. The most important message of the manuscript is the coordination of Cu_A and Cu_B assembly. In this line, Fig 4 shows how the redox state of the Cu_A assembly chaperones SCO1 and SCO2 is modified by the presence or absence of Cu_B assembly chaperones.

2. Some of the statements do not match the data. For example, in line 137-138 the authors state “overexpression of COX19 in the COX11-KO cell line did not improve its residual activity and vice versa,” however there is no data for the “vice versa” part of the statement.

The reviewer is correct. Indeed, the data showing that COX11 overexpression in the COX19-KO cell line did not suppress CIV activity defect were previously not included. This is now fixed and the relevant data are a part of the **Supplementary Fig. 2** (see panels **S2D** and **S2G**).

Similarly, the statement in line 141, even in the presence of exogenous Cu (Fig. 1F) and line 146 (Fig. 1H) there is no indication of exogenous copper in the figure.

Again, the reviewer is correct. The data showing the lack of effect of exogenous copper supplementation on CIV activity in relevant COX11-KO strains were not included. These data have been now added to **Fig. 1G, 1H** and **1J**.

3. The authors should show the levels of COX1 in Fig 1E.

The immunoblot showing COX1 levels has been added to the new **Fig.1K**.

4. The decrease in most of the CcO biogenesis factors observed in COX19KO and PET191KO shown in Supplementary Fig. 2A and B could be simply due to decreased mitochondrial mass. Instead of using Actin as a loading control the authors should use other mitochondrial proteins such as VDAC, an inner mitochondrial membrane protein and a mitochondrial matrix protein. This will tell whether the effect of the loss of COX19 and PET191 is specific to CcO biogenesis factors or is common to other mitochondrial proteins.

We agree with the reviewer and have included VDAC and TOM20 immunoblots as loading controls.

5. Since COX11 has been previously shown to participate in Cu_B site formation, the data showing its more prominent role in Cu_A site formation are really surprising. The author should provide an explanation or more experimental data to support this observation. Is it possible that COX11 function has diverged from unicellular eukaryotes to metazoans? The authors should test whether human COX11 can replace yeast COX11KO or vice versa. The results from this experiment could be more meaningful than sequence-based prediction of COX11 function.

As mentioned at the outset, we are not proposing that COX11 is the copper donor for Cu_A site in COX2. Instead, we show that COX11 forms complexes with COX2 metallochaperones (e.g., SCO1/SCO2), thereby contributing to coordinated metalation of the Cu_A and Cu_B sites. Unexpectedly, however, our data clearly demonstrate that COX11 – as a part of the metallochaperone complex – interacts with the newly-synthesized COX2 prior to interaction with COX1, which allowed us to propose a refined course of events (e.g., Cu_A is formed before Cu_B) during CIV biogenesis.

At any rate, the heterologous complementation experiments proposed by the reviewer are very interesting and have been carried out as requested. As shown in the new **Supplementary Fig. 3**, human COX11 cannot complement the yeast *cox11Δ* mutant. Likewise,

the expression of yeast *COX11* in human *COX11*-KO line is unable to confer respiratory capacity in these cells.

6. Another example of data not matching the text is in Supplementary Fig. 3, where the authors state “PET191....., weakly suppresses the respiratory growth defect of a *COX11* strain” The statement is incorrect as there is no rescue in YPGL respiratory media. There is a slight rescue in SCGal or YPGal fermentative media, though severe growth defect of *COX11* KO in this media is surprising. Is growth defect of *COX11*KO in YPGAL media previously reported? The authors should confirm if this phenotype is indeed a result of *COX11*KO and not due to some other mutation in the strain background by transforming back with *COX11* plasmid and showing full rescue of growth in YPGal.

We agree with the reviewer’s point that the statement about the composition of the yeast culture media was not precise and modified it accordingly.

Thanks to the reviewer’s comment we have realized that a set of growth assays presented in **Supplementary Fig. 3** was mislabeled. Rather than complete YPGal plates, the SC-Gal plates were used in these experiments. The corresponding labels in what is now **Supplementary Fig. 5** have been corrected. Importantly, a partial growth defect on YPGal medium was reported previously², and is expected to be more severe on SC-Gal plates.

7. Figure 1H and Supplementary Figure 2F are identical and one of them should be removed.

Good point. As per reviewer’s suggestion, the old **Supplementary Fig. 2F** panel has been eliminated, and new panels added to an updated **Supplementary Fig. 2G** and **H**.

8. Some of the data are internally inconsistent. For example, there is discrepancy between Fig. 1E and 1F, where *SCO1* overexpression in *COX11*KO results in increased levels of *COX1* (Fig. 1E) but not the CcO activity (Fig. 1F).

We thank the reviewer for this notion. The experiment in question has been repeated several times and we now present the densitometry data, normalized by ACTIN levels, to show that *SCO1* overexpression in *COX11*-KO cells does not appreciably increase the levels of *COX1* or *COX2* (see new **Fig. 1K**).

9. In line 157-158, “mutation abolished PET191 capacity to suppress the *COX11*-KO CcO deficiency” is misleading because even the WT PET191 overexpression failed to rescue *COX11* KO (Supplementary Fig. 4F compare lanes 2 and 4). Also, it is not clear why supplementary figure 4G is cited to describe this result.

Fig. 1D shows that overexpression of WT but not the mutant PET191 can partially rescue Complex IV deficiency in the *COX11*-KO cells. The CcO activity rescue by PET191 does not result in a significant enhancement of free complex IV but is accompanied by a marked increase of the respiratory supercomplexes-incorporated complex IV. These results are now quantified and included as **Supplementary Fig. 4F** and **G**). This respective information has been also included in the text.

10. The graphical description of reverse thiol-trapping approach should be improved. For example, the authors could provide the structures of the products at each step in Fig2A instead of just providing the structure of IAA or AMS. This can be illustrated in the form of three examples depicting the following scenarios: All three cysteines are reduced and free, one free

cysteine and a disulfide, one free cysteine and two copper bound cysteines. Fig 2B is misleading and should be removed.

These are excellent suggestions. The modified schematics to explain our thiol-trapping approach have now been included as a new **Fig. 2A**, and previous Fig 2B has been eliminated.

11. The interpretation of reverse thiol trapping data is misleading. For example, in line 170, “In WT HEK293T cells 50% of COX11 is Cu(I). COX11” Treatment with just AMS (last lane in fig 2C) should give us details about the cysteines that are not oxidized i.e., free and copper bound cysteines should react with AMS. However, in Fig 2C only one cysteine reacted with AMS indicating that the other two are oxidized and not copper bound. In that case, the authors should clarify why they say that in WT, 50% of COX11 is copper bound. Similar problems are seen in fig 2E.

Adding directly AMS to intact mitochondria did not yield consistent results, likely because the bulky AMS molecule is not fully membrane-permeable. Therefore, these data were not used for quantification purposes.

To obtain robust data we have only used the results of experiments from the reverse thiol-trapping, in which the native free cysteines were alkylated with IAA and subsequently treated with TECP and SDS to fully reduce and expose natively oxidized/inaccessible thiols, and then incubated with AMS to derivatize these in vitro-exposed thiols.

12. In lines 172-173 (Fig. 2D) “only apo-COX11 was detected in COA6-KO” the data from lanes 3 and 4 do not add up.

The reviewer is likely confused here for the same reasons as explained for point #11. We have modified the presentation of these data and the description in the text for better clarity.

13. In line 183-184, the statement that “COX11 exhibits a stable interaction with newly synthesized COX2 but not with COX1 in WT HEK293T cells” is not accurate. There is an enrichment of signal for COX1 in HEK293T cells (though it is less than COX2) (Fig. 3A).

We respectfully disagree with the reviewer’s assessment. The experiment in question has been repeated multiple times and we did not observe newly-synthesized COX1 consistently immunoprecipitating with COX11 above the background levels. Additional replicates of these experiments are a part of the **Supplementary Fig. 7A**. The respective data have been thoroughly quantified and is presented in **Fig. 3A**.

14. Supplementary Figure 5A do not show that interaction persists between COX11 and COX2 in the absence of COX1 as stated in line 185.

The reviewer is correct. The data showing that interaction persists between COX11 and COX2 in the absence of COX1 are presented in **Fig. 3A**, and the text has been modified accordingly.

15. The heatmap in Supplementary Fig. 5B do not match the statement in line 192-193.

We thank the reviewer for raising this point. The statement refers to the interaction of COX11 with newly synthesized COX2. The figure panels have now been properly cited as follows: “The interaction persists in the absence of COX1 (**Fig. 3A**), but it is abolished in the absence of

COX20, a factor that stabilizes COX2 during insertion of its N-proximal transmembrane domain, or COX18, which transiently interacts with COX2 to promote translocation across the inner membrane of the COX2's C-tail that harbors the apo-CuA site (**Supplementary Fig. 7B**)."

16. In lines 193 -194, the authors state that SCO1, COA6 and COX16 failed to interact with COX2 in the absence of PET191. In Supplementary Fig 2B, the authors have shown that in PET191-KO the levels of SCO1, COA6 and COX16 are already severely down. So how did the authors account for these decreased levels in the interpretation of their IP data? The authors should also provide the levels of immunoprecipitated protein in Supplementary Fig 5B as controls.

Our statement in the original lines 193-194 referred to the amount of newly synthesized COX2 in the assays reported in **Supplementary Fig. 7**. In panel **7G** we presented the average quantification of immunoprecipitated COX2 relative to the levels of immunoprecipitated protein and expressed as percentage of WT, across the cell lines of interest from three independent experiments. As per reviewer's request, we have now provided data on the levels of immunoprecipitated protein in all panels of **Supplementary Fig. 7**.

17. In lines 196-197 the authors state that the interaction of COX11 with COX2 is independent of PET191 or COX19. However, in Supplementary Fig 5B, the COX11 IP did not show any co-IP of COX2 in WT itself in both cases. In that case, how did the authors come to this conclusion? Also, this contradicts their data in Fig 3A that COX11 interacts with COX2. This further raises questions about the reproducibility of the results from this study.

The original sentences in lines 195-199 were: "The lack of COX2 did not promote COX11-COX1 association, nor did COX19 or PET191 stably interact with newly synthesized COX2 (**Supplementary Fig. 5A**). Instead, the three proteins - COX11, COX19, and PET191- affected the interaction between COX2 and known COX2 folding and CuA metalation chaperones. The absence of COX11 diminished the interaction of COX2 with SCO2 by 20% and of COX20 by 50% (**Supplementary Fig. 5B**)."

The original **Supplementary Fig. 5** is now **Supplementary Fig. 7**. In this updated figure, we are using FLAG epitope-tagged proteins for immunoprecipitation with anti-FLAG-conjugated beads. Therefore, the WT HEK293T cells that are not expressing FLAG-tagged proteins merely serve as negative controls.

The reviewer is correct in their assessment that in the absence of PET191 or COX19, the interaction COX11-COX2 is abolished. To avoid confusion, we have split the first sentence into two as follows: "The lack of COX2 did not promote COX11-COX1 association (**Supplementary Fig. 7B**). Furthermore, COX19 or PET191 did not stably interact with newly synthesized COX2 (**Supplementary Fig. 7A and 7C**)."

18. The authors should explain how they normalized the data in figure 5A.

This is an excellent point that was also raised by the other two reviewers. In the original version, the data in question were expressed as [Cu] in parts per billion normalized by the background value (anti-FLAG beads in WT cells not expressing any FLAG-tagged protein). In the revised manuscript, we present the data as total Cu / S ratios. The specifics of the method have now been detailed in the **Fig. 5** legend as well as the methods section.

19. In lines 297-300 the authors should explain why they say PET191 binds copper in the absence of COX11. ICP-MS of the cross-linked interactome is not a clear way to see if a protein binds copper or not.

These are good points that we have now clarified in the manuscript. We did not use crosslinkers in these assays. Rather, we have only immunoprecipitated the protein interactome that is stable in our extraction conditions. In the presence of COX11, PET191 co-immunoprecipitates with SCO1, COX16 and COX10 (**Fig. 3B** and **raw data**). However, we found that in the absence of COX11, PET191 did not stably interact with any other protein of interest (**raw data**). Therefore, our results suggest that the Cu measured in our assays is bound to PET191.

Minor concerns

1. In lines 285 and 288 Fig.3D should have been Supplementary Fig 6H.

We have cited both figures (**Fig. 3D** and **Supplementary Fig. 10H**) in each case.

2. Each blot in a figure should be considered a separate panel and should be referenced in the text appropriate for ease of understanding - ex: Supplementary Fig 5B can be split into 8 panels.

This is a great suggestion that we have tried to follow by splitting panels as much as possible, including **Supplementary Fig. 7** (formerly Supplementary Fig. 5).

3. What is WT-NT in Fig 5A? Explain in the figure legend

It is intended to refer to WT cells without tagged proteins. However, we agree that this acronym is confusing and have eliminated “NT” from the figure and text.

4. Explain the abbreviations Ex, Un used in Fig 3 in the figure legend.

The text explaining relevant abbreviations has been added.

5. Why is COX1 not included in Fig 3C?

We have now included COX1 in the respective figure, showing that COX1 indeed did not co-purify in any of the IP assays carried out in the absence of crosslinker.

Reviewer #3:

The study by Nývltová investigates the roles of COX11, COX19 and PET191/COA5 in COX assembly in metazoans. The functional characterization of these three factors outside of budding yeast has been lacking, and is therefore a very worthwhile avenue of investigation and one whose advancement is crucial to a fuller understanding of how the holoenzyme is made. An impressive amount of data was collected and is presented but the manuscript in its current construction is very difficult to digest. Moreover, there appears to be somewhat of a disconnect between the emphasis placed on the content of the Title and the Abstract relative to the central findings described by the authors. More specifically, it has been known for some time that stalled assembly intermediates relating to maturation of early assembly modules leads to the production of cytotoxic reactive intermediates (e.g. Khalimonchuk et al. 2007, Veniamin et al. 2011). In contrast the Abstract states the central finding as “Here, we report that in human cells the CcO copper chaperones form macromolecular assemblies and cooperate with several twin-CX9C proteins to control heme a biosynthesis and coordinate copper transfer sequentially to the CuA and CuB sites.”. Although too strongly worded in this Reviewer’s opinion (see below for more detail), this is where the real novelty of

the work under consideration is found. My specific comments, questions and/or criticisms can be found below.

We concur with the reviewer on their notion. Indeed, it has been previously shown in yeast, that stalled CcO assembly leads to the production of cytotoxic reactive intermediates. However, to the best of our knowledge, this situation has not been reported in human cells, and it could be quite different, as happens to be the case for human CcO assembly process. Furthermore, we report on the regulatory mechanism to prevent accumulation of assembly intermediates with COX1 containing heme A cofactor but not copper. This mechanism does not involve translational downregulation of COX1 as it occurs in yeast but leverages the entrapment of heme A biosynthetic enzymes and CcO-specific copper chaperones in what we refer to as metallochaperone complexes. Nonetheless, we agree that the intricacies of CcO assembly process in human cells is the focus of this contribution and therefore have changed the title of the manuscript to **Coordination of metal center biogenesis in human cytochrome c oxidase**, while maintaining the data and important discussion of some of the physiological consequences resulting from the loss of such coordination.

1. Fig 1A/Fig S2A,B: There is no doubt that the KO lines are authentic, and I appreciate how thoroughly they were characterized before downstream experiments were conducted. However, actin is a loading control to ensure equal amounts of whole cell extracts were loaded across lanes and I would like to see a mitochondrial loading control (e.g. TOM20/40) to ensure that organelle content is comparable when comparing WT cells to knockout and functionally complemented lines.

Thank you for raising this point, which was also raised by the Reviewer #1. We agree with the reviewers' assessment and have now added immunoblots of VDAC and TOM20 as loading controls.

2. Fig 1C/Fig S2C,D: What is "sub CIV"? Assembly intermediates have traditionally been viewed as S1-S3, with S2 being a prominent intermediate seen when there is a failure to mature the CuA site of COX2. If S2 is in fact accumulating in COX11 KO (and COX19 KO and PET191 KO) cells, it would be consistent with your other findings that COX11 interacts with COX2 and is crucial for its maturation.

The "sub CIV" designation was intended to refer to the bulk CIV subcomplexes/assembly intermediates accumulating in each cell line. We have followed the reviewer's advice and included traditional S1-S4 labels in most BN-PAGE blots and show that these intermediates contain COX1 but do not COX2 (new **Supplementary Fig. 8D**).

3. Fig 1H/Fig S2F: The finding that PET191 overexpression can partially complement the COX11 KO cell lines and that its Cys residues are required (Fig 1F) is an important one. Why, however, have the authors duplicated identical data in two distinct figure panels? This is worrisome.

We thank the reviewer for noticing this error and apologize for inadvertently including duplicated data. We have removed the original **Supplementary Fig. 2F**.

4. Fig 1I: The relative abundance of heme a/a₃ based on the Soret band argues that COX11 and COX19 are not required for hemylation of COX1, which is a really neat finding. Why,

however, does the peak not return to 603nm upon overexpressing PET191 in the COX11 KO? This suggests that even though you have an increase in enzyme activity and COX2 abundance (Fig 2E, S4E), that the binuclear center may not be in its native state. Consistent with this idea, no complementation of native enzyme levels are observed by BN PAGE (Fig S4F). Reconciling these findings is important yet isn't raised in either the Results or Discussion.

We have now added a paragraph in the results section, extending the description of the data pointed out by the reviewer. Our data show that in addition to enhancing CcO activity, PET191 overexpression also increased the steady-state levels of CcO catalytic subunits COX1 and COX2 (**Fig. 1K**), albeit the native enzyme levels were only increased at the supercomplex level (quantification of multiple experiments is shown in **Supplementary Fig. 4F-G**). Furthermore, PET191 overexpression slightly yet significantly modified the $a+a_3$ cytochrome spectra (**Fig. 1F** and **raw data**), with a fraction of the $a+a_3$ peak red-shifted (~600 nm vs 597nm in *COX11-KO*) suggesting that in this fraction, the binuclear center may be in – or at least closer to – its native state (**Fig. 1F**). All these pieces of information are now discussed in the text as requested by the reviewer.

5. Fig 2: The experiments quantifying the redox state of the cysteinyl sulphurs of COX11 are very convincing. My one question for the authors is how do they discriminate between a direct versus an indirect effect of deleting COX19, PET191 or COA6 on the Cys redox state of COX11? More specifically, given that we know this will lead to the accumulation of stalled assembly intermediates how can the authors rule out the possibility that the disproportionate oxidation of cysteinyl sulphurs in COX11 isn't attributable to enhanced radical production and therefore changes in the local redox environment? It would seem an experiment where COX10 is silenced as was done in Fig 5C would be an important one to discriminate between a direct and indirect effect of loss of these three COX assembly factors on the redox regulation of COX11.

We thank the reviewer for their appreciation of our reverse thiol trapping assays. To clarify, the absence of COX19, PET191 or COA6 does not result in oxidation of the cysteinyl sulfurs in COX11, but in their full reduction (**see Fig. 2**). Nevertheless, we have performed the experiment requested by the reviewer because it could inform on the coordination of heme A biosynthesis and CcO copper center assembly. Our data (**Supplementary Fig. 12**) show that COX10 silencing in WT HEK293T cells alters the redox state of COX11 and SCO1/SCO2 just like in the absence of COX19, PET191 or COA6. Silencing of *COX10* in *PET191-KO* cells did not further modify the redox state of COX11 and SCO1/SCO2. To interpret these data, we must consider that depletion of COX10 steady-state levels attenuated the formation or stability of early metallochaperone complexes containing PET191 and COX11 (**Supplementary Fig. 12**). If these complexes cannot form – just like in the absence of COX10, PET191, or COX19 – the residual apo-COX11 remains in a fully reduced state (i.e., without copper bound and without forming intramolecular disulfides). In WT cells, only when COX11 becomes loaded with copper, the two copper coordinating cysteines would become inaccessible to alkylating agents *in cellulo*, and only accessible when the samples are reduced with DTT and solubilized/unfolded with SDS (see our reverse thiol trapping assays). Therefore, our data indicate that the apparent redox state of COX11 is dependent on the integrity of metallochaperone complexes containing COX10, COX11, SCO1/SCO2, COX19, PET191 and likely some additional factors (see the model in **Fig. 6**). A common denominator of cell lines depleted for any of these factors, is that COX19 levels are not or barely detectable. This observation prompted us to propose that COX19 may directly control the redox state of COX11, by facilitating conformational change in the protein, thereby rendering it competent for copper binding. Our data indicate that COX19 is

released from the maturing CcO assembly intermediates once COX2 and COX1 have been already associated, but not together with COX11, and prior to COX11-mediated copper transfer to COX1; a step that coincides with SURF1 association – presumably to facilitate the assembly of the heme a_3 site.

6. Fig 3C: The finding that COX11 interacts with newly synthesized COX2 is convincing and very novel. This Reviewer is having trouble, however, rationalizing why fewer interacting partners for COX11 and PET191 were identified using MS than IP/WB analysis, given that the former approach is infinitely more sensitive. A gold standard when using physical methods to establish that two proteins interact is a reciprocal IP. Even though this would be an onerous task given the sheer number of IPs conducted in this study, it seems crucial since the manuscript as a whole aims to build a model (Fig 6) that describes how subassembly intermediates are temporally populated with ancillary factors as a given module is matured and merged with others to build the native enzyme.

Several interacting partners of COX11 and PET191 detected by IP/WB analysis in non-crosslinked samples are either small (e.g., all twin-CX₉C proteins) or hydrophobic (e.g., COX1) polypeptides that are notoriously uneasy to detect by MS. Regarding the reciprocal IPs, it is worth mentioning that we have done most if not all of them – or at least all the relevant ones – as presented in **Fig. 3B - D**, and all the **raw data** that were used to generate the heatmaps in **Fig. 3D**. These data have in fact allowed us to build our model presented in **Fig. 6**.

7. Figs 2, 4 and S4: Panels A and B argue that PET191 retains some residual function in the absence of its conserved cysteines. However, based on Fig S4F it acts as a dominant negative when expressed in a COX11 KO background, further reducing residual COX content. Given the pronounced effects of loss of PET191 on the redox state of the cysteinyl sulphurs of COX11 (Fig 2C), SCO1 (Fig 4B) and SCO2 (Fig 4G), all of which become more oxidized, it would seem important to determine whether overexpressing mutant PET191 exerts similar or different redox effects on these three factors. Again, with the current data it is impossible to discriminate between the possibility that PET191 acts directly on COX11, SCO1 and/or SCO2 to modulate their redox state or indirectly by serving as a scaffold protein/stabilizing factor that regulates the activity of another thiol oxidoreductase

The reviewer is right in their assessment that when expressed in WT HEK293T cells, the PET191(C30A, C41A) variant exerted a dominant negative effect on CcO assembly (**Supplementary Fig. 4E-F**). This could be due to the mutant protein partially compromising the import and thus decreasing the mitochondrial levels of the functional WT protein (**Supplementary Fig. 11B**), and/or competing with the WT variant for the interaction with other metallochaperones. That being said, multiple replicates of the experiment and normalized quantification of the immunoblot signals have shown that PET191(C30A, C41A) does not have a dominant negative when expressed in the COX11-KO background (**Supplementary Fig. 4E-F**).

Nonetheless, the reviewer's comment motivated us to further explore the properties of mutant PET191. We have demonstrated that both WT and mutant PET191 interact with the same set of CcO assembly factors. When expressed in PET191-KO cells, these proteins interact with COX11, COX19, COX10, SCO1, SCO2, COA3 and COX16 (**Fig. 3D** and **Supplementary Fig. 4D**); and when expressed in COX11-KO cell, both PET191 variants maintain their interactions with COX10, SCO1, SCO2, COA3 and COX16 (**Supplementary Fig. 4D**). Therefore, our data suggest the existence of an earlier metallochaperone complex

involving these factors (**Supplementary Fig. 11C**). Given the marked effects of PET191 loss on the redox state of the cysteinyl sulfurs of COX11 (**Fig. 2B**), SCO1 (**Fig. 4B**), and SCO2 (**Fig. 4G**) – all of which become more reduced – we examined the effect of the mutant PET191 overexpression on the redox state of three factors. PET191(C30A, C41A) overexpression in HEK293T WT cells resulted in more oxidized COX11, whereas in *PET191*-KO cells the expression of this PET191 variant rendered COX11, SCO1, and SCO2 more oxidized - to the levels closer or equal than in WT cells (**Supplementary Fig. 11C-E**). Notwithstanding, the restoration of CcO assembly by PET191(C30A, C41A) is only partial in that case, which could be due to an increased resistance towards the release from the apo-metallochaperone complex and substitution by COX17.

Together with the data showing that absence of COX10 disrupts the metallochaperone complexes, drastically decreases the levels of PET191, and affects the redox state of COX11 and SCO1 (new **Supplementary Fig. 12A-E**), these results strongly suggest that the integrity of the metallochaperone complexes is a key determinant of the redox state of their components. Whether or not PET191 plays a direct role remains to be determined by future studies. However, at least for COX11, we favor a model wherein upon the release of PET191 from the metallochaperone complex, the subsequent incorporation of COX17 facilitates metalation of COX11 and changes in its apparent redox state as seen in our thiol trapping assays. All this information has been now included in the text.

8. Fig 5A: The figure legend says the copper content of the metallochaperone modules was quantified yet the methods describe that the analysis was conducted on whole mitochondria. If the latter is in fact what was used as starting material, these results cannot discriminate where the copper itself is found. For e.g., it could be part of the labile pool housed in the matrix, it could be bound to metallochaperones or to target proteins (COX1, COX2). This Reviewer only sees a significant increase in copper upon overexpressing COX11 or PET191 with all other conditions being comparable to WT NT. No data is shown for COX11 KO or PET191 KO cells. Thus, there is a limited ability to draw firm conclusions from this analysis, especially since the authors are trying to temporally link hemylation and copper insertion to specific stages of holoenzyme assembly. A more arduous but rigorous approach would be to IP the MCC and earlier subassembly intermediates (Fig S5D argues they can do this), fractionate them by size exclusion under native conditions using an HPLC that is coupled to an in-line MS spectrometer (e.g. Soma et al. 2019). This would allow you to definitively link a given metal to a specific stage of holoenzyme biogenesis.

We thank the reviewer for pointing this out and apologize for the mix-up. The former **Fig. 5A** (now **Fig. 5D**) shows copper content in COX11- and PET191-containing metallochaperone modules (i.e., protein complexes) measured by inductively coupled plasma mass spectrometry (ICP-MS). For these experiments, mitochondria were purified from the *COX11*-KO or *PET191*-KO cells expressing FLAG epitope-tagged COX11 or PET191 proteins (WT or mutant variants thereof). As suggested by the reviewer, the proteins were immunoprecipitated using anti-FLAG conjugated beads and their metal content assessed by ICP-MS. In the original version of this manuscript, the data were expressed as [Cu] in parts per billion normalized by the background value (anti-FLAG beads in WT cells not expressing any FLAG-tagged protein). Now, as per reviewers' #1 and #2 request, we have expressed the data as total Cu normalized by [³⁴S]. The specifics of the method are now schematically presented in **Fig. 5C** and included in the **Fig. 5** legend as well as the methods section.

9. The Discussion: The real novelty of this manuscript is its advancement of our understanding of COX11, COX19 and PET191/COA5 function. Yet many holes remain in that knowledge.

Thus, this Reviewer found the Discussion to be focused too much on building a holistic model of holoenzyme assembly that overstretched the interpretation of the data in hand with respect to when and by whom the copper sites are formed to allow modules to merge and holoenzyme assembly to proceed. While not much is known about COX11, COX19 and PET191/COA5 function in metazoans, some uncited studies have looked at these proteins and how they may interact with other COX assembly factors to promote holoenzyme assembly. A significant rewrite is warranted.

We agree with the reviewer on the point that the data presented in our manuscript advances the understanding of the roles of COX11, COX19 and PET191 in CcO biogenesis. However, we respectfully disagree on the notion that this is the “real” (if the reviewer implies “the only”) novelty. We believe that our findings on the coordination of CuA and CuB assembly, the formation of metallochaperone complexes, and the role that the three proteins in question play in these processes are novel and fully justify our refined CcO biogenesis model. We have intended to cite all the available relevant literature and will gladly include any relevant citations that the reviewer may find missing. The very constructive criticism provided by the three reviewers has helped us to solidify our previous conclusions and significantly modify the Discussion section. We therefore believe that with the reviewers’ valuable input our working model is now sounder. Furthermore, this model provides a framework for future studies aiming to close the remaining gaps in knowledge on CcO biogenesis in metazoans.

MINOR:

1. Fig S3D needs better labelling and/or a clearer description in the legend given that at present the distinction between the top pair and bottom pair of graphs is hard to understand.

Thank you for pointing this out. To avoid confusion, we have now separated the two sets of data into two different panels (now presented as **Supplementary Fig. 5D-E**) demonstrating sensitivity of the indicated strains to hydrogen peroxide. The dominant negative effect of the *cox11*Δ mutation is shown in panel **5D**. The dominant positive effect of the *pet117*Δ mutation is shown in panel **5E**.

2. Pg 10 “The entrapment of COX10 in complexes with CuA and CuB chaperones until their Cu loading reflects a concerted regulation of metalation of all the redox centers in CcO.” I cannot find any direct evidence that is presented to tie it metallation, particularly of the copper centers.

Our data indicate that COX10 (and COX15 too) form complexes with the copper metallochaperones COX11, SCO1, SCO2, and COX17, from which COX10 is released together with COX17 (see the model in **Fig. 6**). We assume that COX17 is released once it has transferred copper to COX11, and SCO1/SCO2. We realize that this is rather indirect evidence, and therefore modified the sentence highlighted by the reviewer - the verb “reflects” has been substituted with “suggests”.

3. Pg 13 “We find that contrary to the established dogma, COX11 is important but dispensable in metazoans.” This statement is very strong given the finding was generated in a model human cell line cultured under supraphysiological conditions and suggests that humans lacking functional COX11 would be free of disease.

We concur with the reviewer’s notion and changed the sentence to: “We find that contrary to the established dogma, although COX11 is important, a COX11-independent route can contribute up to 15% of assembled and functional CcO”.

References

1. Thompson, A.K. *et al.* Mutagenic analysis of Cox11 of *Rhodobacter sphaeroides*: insights into the assembly of Cu_B of cytochrome *c* oxidase. *Biochemistry* **49**, 5651-5661 (2010).
2. Bode, M. *et al.* Redox-regulated dynamic interplay between Cox19 and the copper-binding protein Cox11 in the intermembrane space of mitochondria facilitates biogenesis of cytochrome *c* oxidase. *Mol. Biol. Cell.* **26**, 2385-2401 (2015).

REVIEWERS' COMMENTS

Reviewer #1 (Remarks to the Author):

Thank you to the authors for diligently addressing all of my comments and concerns. I believe these have all been sufficiently addressed and I appreciate the authors' hard work to do so.

Reviewer #2 (Remarks to the Author):

The authors have performed additional experiments and have corrected many errors from their previous draft. However, some of the results from new experiments do not support the main conclusion from the paper that biogenesis of CuA and CuB sites are coordinately regulated. This is because the function of COX11, which has been previously shown to act as a metallochaperone for CuB site formation in yeast, is not conserved in metazoans. It is now not clear which protein is responsible for inserting copper in CuB site in metazoans? Since the premise for the conclusion that CuA and CuB site formation is coordinately regulated comes from the assumption that human COX11 has the same function as yeast Cox11 in making in CuB site, the new results now does not support this claim. Additionally, the abstract of the paper is very vague and misleading (only two twin Cx9C proteins – COX19 and PET191 have been studied and not “several twin Cx9C proteins” as stated in the abstract).

Reviewer #3 (Remarks to the Author):

The authors have addressed all of the concerns I raised in my original review. I congratulate them on the completion of a very thorough and well constructed study.

Response to Reviewers

Reviewer #2:

The authors have performed additional experiments and have corrected many errors from their previous draft. However, some of the results from new experiments do not support the main conclusion from the paper that biogenesis of CuA and CuB sites are coordinately regulated. This is because the function of COX11, which has been previously shown to act as a metallochaperone for CuB site formation in yeast, is not conserved in metazoans. It is now not clear which protein is responsible for inserting copper in CuB site in metazoans? Since the premise for the conclusion that CuA and CuB site formation is coordinately regulated comes from the assumption that human COX11 has the same function as yeast Cox11 in making in CuB site, the new results now does not support this claim.

We respectfully disagree with the opinion that the role of COX11 is not conserved in metazoans. I can only speculate that the reviewer reaches this conclusion based on the fact that yeast Cox11 and human COX11 are not able to function in the heterologous context. In any heterologous complementation assay, only a positive result is conclusive. If the result is negative, it does not necessarily mean that the yeast and human homolog proteins have different functions. Considering the large number of dynamic protein-protein interactions in which COX11 is involved, multiple proteins may have co-evolved still retaining the same main function. Of course, in more complex organisms, additional regulatory functions may have been acquired to adapt to the biological context.

As a reference, over the last 20 years, we have performed yeast-human heterologous complementation assays for most if not all cytochrome c oxidase assembly factors. Only human COX10 was able to substitute, very partially, for yeast Cox10 in a *cox10* deletion strain (Valnot et al, Hum Mol Genet. 2000 May 1;9(8):1245-9). Regarding yeast and human COX11, both proteins undergo an intimate physical and functional interaction with COX19. We show that human COX11 binds copper and that this capacity is attenuated in the absence of COX19. We do not have any data supporting a fundamental role for human and yeast COX11 other than copper binding and delivery to COX1, as it was demonstrated in bacteria.

Additionally, the abstract of the paper is very vague and misleading (only two twin Cx9C proteins – COX19 and PET191 have been studied and not “several twin Cx9C proteins” as stated in the abstract).

Concerning the abstract, the two twin CX9C proteins best characterized in our paper are COX19 and PET191, as mentioned by the reviewer. However, additional proteins from this family, including CMC1 and COA6, were also found to undergo dynamic protein-protein interactions very relevant to our study. For this reason, we used the wording “several proteins” in the abstract.